# GPT-ST: Generative Pre-Training of Spatio-Temporal Graph Neural Networks

**Zhonghang Li**[1,2]    **Lianghao Xia**[2]    **Yong Xu**[1,3,4]    **Chao Huang**[2]*

[1] South China University of Technology, [2] University of Hong Kong,
[3] PAZHOU LAB, [4] Guangdong Key Lab of Communication and Computer Network

{bjdwh.zzh,chaohuang75}@gmail.com   aka_xia@foxmail.com   yxu@scut.edu.cn

## Abstract

In recent years, there has been a rapid development of spatio-temporal prediction techniques in response to the increasing demands of traffic management and travel planning. While advanced end-to-end models have achieved notable success in improving predictive performance, their integration and expansion pose significant challenges. This work aims to address these challenges by introducing a spatio-temporal pre-training framework that seamlessly integrates with downstream baselines and enhances their performance. The framework is built upon two key designs: (i) We propose a spatio-temporal mask autoencoder as a pre-training model for learning spatio-temporal dependencies. The model incorporates customized parameter learners and hierarchical spatial pattern encoding networks. These modules are specifically designed to capture spatio-temporal customized representations and intra- and inter-cluster region semantic relationships, which have often been neglected in existing approaches. (ii) We introduce an adaptive mask strategy as part of the pre-training mechanism. This strategy guides the mask autoencoder in learning robust spatio-temporal representations and facilitates the modeling of different relationships, ranging from intra-cluster to inter-cluster, in an easy-to-hard training manner. Extensive experiments conducted on representative benchmarks demonstrate the effectiveness of our proposed method. We have made our model implementation publicly available at `https://github.com/HKUDS/GPT-ST`.

## 1 Introduction

Intelligent Transportation Systems (ITS) is a rapidly growing research field driven by advancements in sensors and communications technology [1, 12]. Spatio-Temporal (ST) prediction, including traffic flow and ride demand, aims to forecast future trends, facilitating decision-making in traffic management and risk response [43], and enhancing people's lives with future travel plans [6]. Deep learning methods, such as RNNs and CNNs, have been widely adopted to model temporal and spatial correlations [5, 51, 50, 45]. Recently, GNNs have gained attention for spatial and temporal prediction tasks, achieving remarkable results. Initially, researchers built adjacency matrices based on region attributes, such as distance [48, 54]. Subsequent studies introduced learnable graph structures to capture spatial correlations [40, 39], while others focused on dynamic relationships between regions, proposing methods for constructing dynamic graphs [14, 40]. These advancements have significantly contributed to the development of spatio-temporal prediction techniques.

Despite the remarkable results achieved by existing methods in spatio-temporal prediction, there are several issues that remain inadequately addressed. **i) Lack of customized representations of specific spatio-temporal patterns.** Customization can be categorized into two key aspects: ***time-dynamic*** and ***node-specific*** properties in both the time and spatial domains. Time-dynamic patterns exhibit variations across different time periods, such as contrasting patterns between weekends and

---

*Corresponding author.

37th Conference on Neural Information Processing Systems (NeurIPS 2023).

weekdays within the same region. Moreover, correlations between regions dynamically evolve over time, surpassing the capabilities of static graph representations. Node-specific patterns highlight the distinct time series observed in different regions, rather than a shared pattern. Additionally, it is important to ensure that different regions retain their individual node characteristics even after message aggregation, to prevent interference from prominent nodes in the spatial domain. We believe that encoding all of these customized characteristics is essential to guarantee the model robustness. However, existing works often fall short in providing a comprehensive consideration of these factors. **ii) Insufficient consideration of spatial dependencies at different levels.** Most approaches primarily focus on pairwise associations between regions when modeling spatial dependency, but they overlook the semantic relevance at different spatial levels. In the real world, regions with similar functions tend to exhibit similar spatio-temporal patterns. By performing clustering analysis on different regions, the model can explore common characteristics among similar regions, thereby facilitating improved spatial representation learning. Moreover, there is a lack of adequate modeling of high-level regional relationships across time in current studies. Spatio-temporal patterns between different types of high-level regions may exhibit dynamic transfer relationships. For instance, during work hours, there is a notable movement of people from residential areas to work areas, as illustrated in the right part of Figure 1. In this context, changes in people flow within residential areas can provide valuable auxiliary signals for predicting people flow in work areas. It highlights the importance of incorporating both fine-grained and coarse-grained correlations among different levels of regions in order to enhance the predictive capabilities of spatio-temporal prediction models.

An intuitive approach to address the aforementioned challenges is to develop an end-to-end model. However, the current models face difficulties in benefiting from such an approach, as each module in the state-of-the-art (SOTA) model is intricately refined, and any disassembly and integration may lead to a degradation in prediction performance. So, *is there a strategy that can allow existing spatio-temporal methods to leverage the advantages of an end-to-end model?* The answer is affirmative. Recent pre-training frameworks, such as ChatGPT [3, 27] in the field of natural language processing (NLP) and MAE [15] in computer vision (CV), have been proposed by pioneers and widely studied in their respective domains. These pre-training architectures construct unsupervised training tasks, involving masking or other techniques, to learn better representations and improve

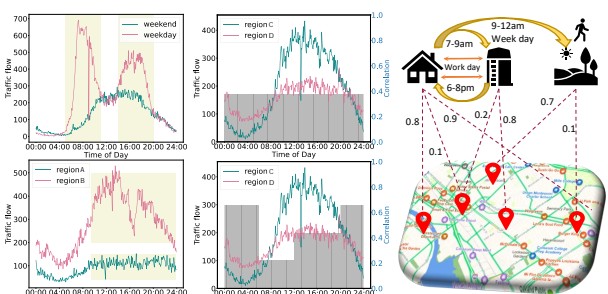

Figure 1: The motivations behind GPT-ST. The upper left figure demonstrates that traffic patterns in the same region exhibit variations across different time periods. Similarly, the lower left figure depicts how this variation can occur between different regions during the same time period. In contrast, existing works typically construct static graphs, failing to capture the dynamic nature of region relationships over time, as shown in the middle figures (lower), where region relationships change dynamically. Lastly, the right figure illustrates the commuting phenomenon between regions belonging to different categories.

downstream task performance. However, the application of such scalable pre-training models in the field of spatio-temporal prediction has been limited thus far. To address the aforementioned challenges, we propose a novel framework called Generative Pre-Training for Spatio-Temporal prediction (GPT-ST). This framework is designed to seamlessly integrate into existing spatio-temporal prediction models, enhancing their performance. Our contributions are summarized as follows:

- We present GPT-ST, a novel pre-training framework designed for spatio-temporal prediction. This framework seamlessly integrates into existing spatio-temporal neural networks, resulting in performance enhancements. Our approach combines a model parameter customization scheme with a self-supervised masking autoencoding, enabling effective spatio-temporal pre-training.

- GPT-ST ingeniously utilizes a hierarchical hypergraph structure to capture spatial dependencies at different levels from a global perspective. Through collaboration with a thoughtfully designed adaptive mask strategy, the model acquires the capability to model both intra-cluster and inter-cluster spatial relations among regions, thereby generating robust spatio-temporal representations.

- We conduct extensive experiments on real-world datasets, and the demonstrated improvement in the performance of diverse downstream baselines showcases the superior performance of GPT-ST.

## 2  Related Work

**Deep Spatio-temporal Prediction Techniques.**  Numerous studies have focused on designing deep neural networks for spatio-temporal data prediction. To capture temporal dependencies, D-LSTM [49] and GWN [40] have developed predictive frameworks based on Recurrent Neural Networks (RNNs) and Temporal Convolutional Networks (TCNs), respectively. GMAN [55] and STGNN [36] have utilized temporal self-attention networks to facilitate long-range temporal learning. In spatial relation learning, DMVST-Net [44] adopted convolutional network to model region associations. STGCN [48] and DCRNN [23] exploited Graph Neural Networks (GNNs) for message passing between regions. STAN [26] consider the interrelationships between regions based on Graph Attention Networks (GAT). Temporal graph networks serve as a similar research baseline aimed at reasoning about dynamic graph structures, such as CAWs [37] and PINT [33]. More recently, some works have explored the combination of ST models with advanced methods, such as Neural-ODE-based [10, 18] and self-supervised learning-based [24], for better spatial and temporal dependencies modelling. Compared to these works, our proposed GPT-ST framework aims at empower diverse deep spati-temporal models with the pre-training stage.

**Pre-trainning methods.** Pre-training models have made significant advancements in recent years, with notable examples including BERT [7, 28], vision transformers [9, 8], masked autoencoders [15, 20], and language models [3, 27, 34]. The success of the autoencoder-based approach, exemplified by MAE, has demonstrated the powerful expressive capability of the mask-reconstruction pre-training framework.  Building upon this, STEP [31] proposed a pre-training model specifically for time series prediction. In other domains, CMSF [41] introduced a master-slave framework for identifying urban villages. The master model pre-trains region representations, while the slave model fine-tunes specific regions for accurate identification.  Researchers have also recognized the need for more efficient mask mechanisms than random masking. AttMask [19] and SemMAE [21] leverage attention mechanisms and semantic information, respectively, to guide the mask operation.  AdaMAE [2] proposes adaptively selecting visible signals with greater reconstruction loss, as they contain more information, and using a higher proportion of masking strategy accordingly. Despite the significant progress of pre-training models in the fields of natural language processing (NLP) and computer vision (CV), their application to spatio-temporal prediction remains relatively limited. Additionally, computational efficiency has been a major bottleneck for transformer architectures in general.

**Self-supervised learning methods for GNNs.** In recent years, self-supervised learning methods for graph data have gained significant attention [53, 38]. Contrastive learning-based GNNs generate different views of the original graph through data augmentation techniques [42, 29]. A loss function is then employed to maximize the consistency of positive sample pairs while minimizing the consistency of negative sample pairs across views. For example, GraphCL [47] generates two views of the graph by applying node dropping and edge shuffling, and performs contrastive learning between them. Another research direction focuses on generative graph neural networks, where the graph data itself serves as a natural supervisory signal for representation learning through reconstruction.  GPT-GNN [17] conducts pre-training by reconstructing graph features and edges, while GraphMAE [16] utilizes node feature masking in both the graph encoder and decoder to reconstruct features and learn graph representations. However, spatio-temporal forecasting tasks require simultaneous consideration of complex temporal evolution patterns and spatial correlation mechanisms. Pre-training paradigms specifically designed for such tasks are still an area of exploration and research.

## 3  Preliminaries

**Spatio-Temporal Data X**. To capture the ST information, we represent it as a three-way tensor $\mathbf{X} \in \mathbb{R}^{R \times T \times F}$, where $R$, $T$, and $F$ correspond to the numbers of regions, time slots, and feature dimensions, respectively. Each entry $\mathbf{X}_{r,t,f}$ is the $f$-th feature of the $r$-th region in the $t$-th time slot.

**Spatio-Temporal Hypergraphs**. We employ ST hypergraphs for ST modeling.  A hypergraph $\mathcal{H} = \{\mathcal{V}, \mathcal{E}, \mathbf{H}\}$ is composed of three parts: **i)** Vertices $\mathcal{V} = \{v_{r,t} : r \in R, t \in T\}$, each of which represents a region $r$ in a specific time slot $t$. **ii)** $H$ hyperedges $\mathcal{E} = \{e_1, ..., e_H\}$, each of which connects multiple vertices to reflect the multipartite region-wise relations (*e.g.*, all residential regions can be connected by one hyperedge). **iii)** The vertex-hyperedge connections $\mathbf{H} \in \mathbb{R}^{N \times H}$, where $N$ denotes the number of vertices. To fully excavate the potential of hypergraphs in region-wise relation learning, we adopt a learnable hypergraph scheme where $\mathbf{H}$ is derived from trainable parameters.

**Spatio-Temporal Pre-training Paradigm**. Our GPT-ST framework aims at developing a pre-trained ST representation method that facilitates the accuracy on downstream ST prediction tasks such as traffic volume forecasting. As depicted in Figure 2, the workflow of GPT-ST can be divided into the *pre-training stage* and the *downstream task stage*. We formulate the two stages as follows:

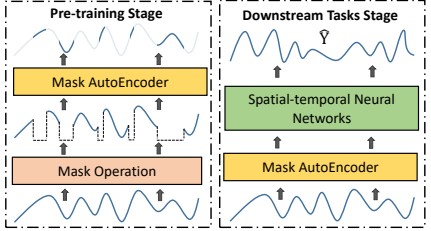

Figure 2: Overall workflow of GPT-ST.

i) The *pre-training stage* of our GPT-ST framework adopts the masked autoencoding (MAE) task as the training objective. The goal of this MAE task is to reconstruct masked information of the spatio-temporal data according to the unmasked ones by learning a ST representation function $f$. For the training ST data during the period of $[t_{K-L+1}, t_K]$, we aim to minimize the following objective:

$$\mathcal{L}\left((1-\mathbf{M}) \odot \mathbf{X}_{t_{K-L+1}:t_K}, \mathbf{W} \cdot f(\mathbf{M} \odot \mathbf{X}_{t_{K-L+1}:t_K})\right) \tag{1}$$

where $\mathcal{L}$ denotes the measure of prediction deviation, $\mathbf{M} \in \{0,1\}$ represents the mask tensor, and $\odot$ denotes the element-wise multiplication operation. $\mathbf{W}$ denotes a predictive linear layer.

ii) After the pre-training stage, the results of our GPT-ST are then utilized in the *downstream task stage*. The pre-trained model yields high-quality ST representations to facilitate the downstream prediction tasks such as traffic forecasting. Specifically, the downstream stage is formulated as:

$$\boldsymbol{\zeta} = f(\mathbf{X}_{t_{K-L+1}:t_K}); \quad \hat{\mathbf{X}}_{t_{K+1}:t_{K+P}} = g(\boldsymbol{\zeta}, \mathbf{X}_{t_{K-L+1}:t_K}) \tag{2}$$

where $\boldsymbol{\zeta}$ refers to the representation generated by $f$ according to the historical spatio-temporal data from the previous $L$ time slots before the $K$-th time slot. The output is the predictions for the next $P$ time slots. Various existing spatio-temporal neural networks can serve as the prediction function $g$.

## 4 Methodology

This section introduces the technical details of the proposed ST pre-training framework GPT-ST. It develops a customized temporal encoder and a hierarchical spatial encoder to obtain large modeling capacity for the pre-trained model. An adaptive mask strategy is proposed to facilitate effective MAE training. Figure 3 demonstrates the model design of our pre-trained spatio-temporal encoding model.

### 4.1 Customized Temporal Pattern Encoding

**Initial Embedding Layer**. We begin by constructing an embedding layer to initialize the representation of the spatio-temporal data $\mathbf{X}$. The original data undergoes normalization using the Z-Score function [1, 39] and is then masked using the mask operation. Subsequently, a linear transformation is applied to augment the representation as $\mathbf{E}_{r,t} = \mathbf{M}_{r,t} \odot \bar{\mathbf{X}}_{r,t} \cdot \mathbf{E}_0$, where $\mathbf{E}_{r,t} \in \mathbb{R}^d$, $\mathbf{M}_{r,t} \in \mathbb{R}^F$, and $\bar{\mathbf{X}}_{r,t} \in \mathbb{R}^F$ represent the representation, mask operation, and normalized spatio-temporal data for the $r$-th region in the $t$-th time slot. The variable $d$ denotes the number of hidden units. Additionally, $\mathbf{E}_0 \in \mathbb{R}^{F \times d}$ represents the learnable embedding vectors for the $F$ feature dimensions.

**Temporal Hypergraph Neural Network**. To facilitate global relation learning, we employ the hypergraph neural network for the temporal pattern encoding, specifically as follows:

$$\boldsymbol{\Gamma}_t = \sigma(\bar{\mathbf{E}}_t \cdot \mathbf{W}_t + \mathbf{b}_t + \mathbf{E}_t); \quad \bar{\mathbf{E}}_r = \text{HyperPropagate}(\mathbf{E}_r) = \sigma(\mathbf{H}_r^\top \cdot \sigma(\mathbf{H}_r \cdot \mathbf{E}_r)) \tag{3}$$

where $\boldsymbol{\Gamma}_t, \bar{\mathbf{E}}_t, \mathbf{E}_t \in \mathbb{R}^{R \times d}$ denote the result, intermediate, and initial region embeddings for the $t$-th time slot, respectively. $\mathbf{W}_t \in \mathbb{R}^{d \times d}$, $\mathbf{b}_t \in \mathbb{R}^d$ represent the $t$-th time slot-specific parameters. $\sigma(\cdot)$ denotes the LeakyReLU activation. The intermediate embeddings $\bar{\mathbf{E}} \in \mathbb{R}^{R \times T \times d}$ are calculated by the hypergraph information propagation. It employs the region-specific hypergraph $\mathbf{H}_r \in \mathbb{R}^{H_T \times T}$ to propagate the temporal embeddings $\mathbf{E}_r \in \mathbb{R}^{T \times d}$ for the $r$-th region along the connections between time slots and $H_T$ hyperedges, so as to capture the multipartite relations among time periods.

**Customized Parameter Learner**. To characterize the diversity of temporal patterns, our temporal encoder conducts model parameter customization for both different regions and different time periods. Specifically, the aforementioned time-specific parameters $\mathbf{W}_t, \mathbf{b}_t$ and the region-specific hypergraph

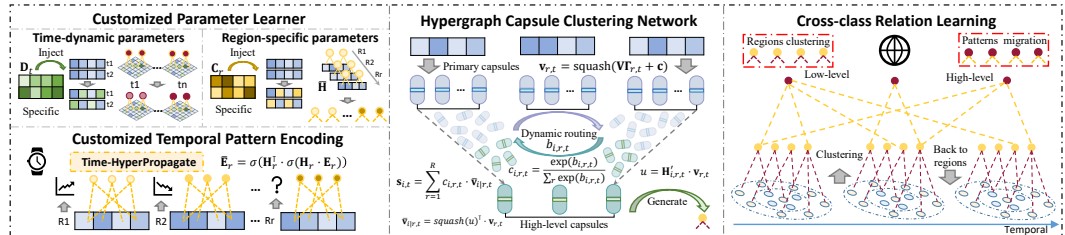

Figure 3: The detailed framework of the proposed GPT-ST.

parameters $\mathbf{H}_r$ are generated through a learnable process, rather than directly utilizing independent parameters. Specifically, the customized parameters are learned as follows:

$$\mathbf{H}_r = \mathbf{c}_r^\top \bar{\mathbf{H}}; \quad \mathbf{W}_t = \mathbf{d}_t^\top \bar{\mathbf{W}}; \quad \mathbf{b}_t = \mathbf{d}_t^\top \bar{\mathbf{b}}; \quad \mathbf{d}_t = \text{MLP}(\bar{\mathbf{z}}_t^{(d)}\mathbf{e}_1 + \bar{\mathbf{z}}_t^{(w)}\mathbf{e}_2) \quad (4)$$

where $\bar{\mathbf{H}} \in \mathbb{R}^{d' \times H_T \times T}$, $\bar{\mathbf{W}} \in \mathbb{R}^{d' \times d \times d}$, $\bar{\mathbf{b}} \in \mathbb{R}^{d' \times d}$ are independent parameters for the three generated parameters, respectively. $\mathbf{c}_r, \mathbf{d}_t \in \mathbb{R}^{d'}$ denote representations for the $r$-th region and the $t$-th time slot, respectively. $\mathbf{c}_r$ for $r \in R$ is a free-form parameter, while $\mathbf{d}_t$ for $t \in T$ is calculated from the normalized time-of-day features $\bar{\mathbf{z}}_t^{(d)}$ and day-of-week features $\bar{\mathbf{z}}_t^{(w)}$. $\mathbf{e}_1, \mathbf{e}_2 \in \mathbb{R}^{d'}$ are their corresponding learnable embeddings. This parameter learner realizes spatial and temporal customization by generating parameters according to the features of specific time slots and regions.

## 4.2 Hierarchical Spatial Pattern Encoding

### 4.2.1 Hypergraph Capsule Clustering Network

Current spatial encoders primarily focus on capturing local adjacent region-wise relations while disregarding the widespread similarities that exist between distant regions. For example, business districts that are geographically apart can still exhibit similar ST patterns. In light of this, our GPT-ST introduces a hypergraph capsule clustering network to capture global region-wise similarities. This network explicitly learns multiple cluster centers as hyperedges, characterizing the global region-wise similarities. To further enhance the hypergraph structure learning, we incorporate the dynamic routing mechanism of capsule network. This mechanism iteratively updates hyperedge representations and region-hyperedge connections based on semantic similarities. As a result, it improves the clustering capability of hyperedges and facilitates the global modeling of dependencies among regions.

Specifically, we firstly obtain a normalized region embedding $\mathbf{v}_{r,t} \in \mathbb{R}^d$ for each region $r$ in time slot $t$, using the previous embedding $\mathbf{\Gamma}_{r,t}$ and the squash function [30]. This embedding is then used to calculate the transferred information $\bar{\mathbf{v}}_{i|r,t} \in \mathbb{R}^d$ from each region $r$ to each cluster center (hyperedge) $i$, within the $t$-th time slot. Formally, these two variables are calculated by:

$$\mathbf{v}_{r,t} = \text{squash}(\mathbf{V}\mathbf{\Gamma}_{r,t} + \mathbf{c}); \quad \bar{\mathbf{v}}_{i|r,t} = \text{squash}(\mathbf{H}'_{i,r,t}\mathbf{v}_{r,t}) \odot \mathbf{v}_{r,t}; \quad \text{squash}(\mathbf{x}) = \frac{\|\mathbf{x}\|^2}{1 + \|\mathbf{x}\|^2} \frac{\mathbf{x}}{\|\mathbf{x}\|} \quad (5)$$

where $\mathbf{V} \in \mathbb{R}^{d \times d}$ and $\mathbf{c} \in \mathbb{R}^d$ are free-form learnable parameters. The hypergraph connectivity matrix $\mathbf{H}'_t \in \mathbb{R}^{H_S \times R}$ records the relations between $R$ regions and $H_S$ hyperedges as the cluster centroids. It is tailored to the $t$-th time slot using the aforementioned customized parameter learner, specifically by $\mathbf{H}'_t = \text{softmax}(\mathbf{d}'^\top_t \bar{\mathbf{H}}')$. Here, $\mathbf{d}'_t$ and $\bar{\mathbf{H}}'$ are temporal features and hypergraph embeddings.

**Iterative Hypergraph Structure Learning**. With the initialized region embeddings $\mathbf{v}_{r,t}$ and hypergraph connection embeddings $\bar{\mathbf{v}}_{i|r,t}$, we follow the dynamic routing mechanism of capsule network to enhance the clustering effect for the hyperedges. The $j$-th iteration is described as follows:

$$\mathbf{s}_{i,t}^j = \sum_{r=1}^R c_{i,r,t}^j \bar{\mathbf{v}}_{i|r,t}; \quad c_{i,r,t}^j = \frac{\exp(b_{i,r,t}^j)}{\sum_r \exp(b_{i,r,t}^j)}; \quad b_{i,r,t}^j = b_{i,r,t}^j + \mathbf{v}_{r,t}^\top \text{squash}(\mathbf{s}_{i,t}^{j-1}) \quad (6)$$

where $\mathbf{s}_{i,t} \in \mathbb{R}^d$ represents the iterative hyperedge embedding. It is calculated utilizing the iterative hyperedge-region weights $c_{i,r,t} \in \mathbb{R}$. The weight $c_{i,r,t}$ is in turn calculated from the hyperedge embeddings $\mathbf{s}_{i,t}$ from the last iteration. With this iterative process, the relation scores and the hyperedge representations are mutually adjusted by each other, to better reflect the semantic similarities between regions and the spatial cluster centroids represented by hyperedges.

After the iterations of the dynamic routing algorithm, to make use of both $b_{i,r,t}$ and $\mathbf{H}'_{i,r,t}$ for better region-hyperedge relation learning, GPT-ST combines the two sets of weights and generates the final embeddings $\bar{\mathbf{s}}_{i,t} \in \mathbb{R}^d$. We begin by replacing $b_{i,r,t}$ with $(b_{i,r,t} + \mathbf{H}'_{i,r,t})$ to obtain a new weight vector $\bar{c}_{i,r,t} \in \mathbb{R}$, and then utilize $\bar{c}_{i,r,t} \in \mathbb{R}$ to calculate the final embeddings $\bar{\mathbf{s}}_{i,t}$.

### 4.2.2 Cross-Cluster Relation Learning

With the clustered embeddings $\bar{\mathbf{s}}_{i,t}$, we propose to model the inter-cluster relations via a high-level hypergraph neural network. Specifically, the refined cluster embeddings $\hat{\mathbf{S}} \in \mathbb{R}^{H_S \times T \times d}$ are calculated by message passing between the $H_S$ cluster centroids and $H_M$ high-level hyperedges as follows:

$$\hat{\mathbf{S}} = \text{HyperPropagate}(\tilde{\mathbf{S}}) = \text{squash}(\sigma(\mathbf{H}^{''\top} \cdot \sigma(\mathbf{H}'' \cdot \tilde{\mathbf{S}})) + \tilde{\mathbf{S}}) \tag{7}$$

where $\tilde{\mathbf{S}} \in \mathbb{R}^{H_S T \times d}$ denotes the reshaped embedding matrix obtained from $\bar{\mathbf{s}}_{i,t}$ for $i \in H_S$ and $t \in T$. $\mathbf{H}'' \in \mathbb{R}^{H_M \times H_S T}$ denotes the high-level hypergraph structure, which is obtained through the aforementioned personalized parameter learner. This parameter customization aggregates the temporal features $\bar{\mathbf{z}}_t^{(d)}, \bar{\mathbf{z}}_t^{(w)}$ for all $t \in T$ as the input, and generates the parameters as in Eq 4.

After refining the cluster representations $\hat{\mathbf{s}}_{i,t} \in \mathbb{R}^d$ for $i \in H_S, t \in T$, we propagate the clustered embeddings back to the regional embeddings with the low-level hypergraph structure, as follows:

$$\boldsymbol{\Psi}_{r,t} = \sigma(\sum_{i=1}^{H_S} c_{i,r,t} \cdot \hat{\mathbf{s}}_{i,t} \mathbf{W}''_r + \mathbf{b}''_r + \boldsymbol{\Gamma}_{r,t}) \tag{8}$$

where $\boldsymbol{\Psi}_{r,t} \in \mathbb{R}^d$ denotes the new region embedding for the $r$-th region in the $t$-th time slot. $c_{i,r,t} \in \mathbb{R}$ denotes the weight for the low-level hypergraph capsule network. $\mathbf{W}''_r \in \mathbb{R}^{d \times d}$ and $\mathbf{b}''_r \in \mathbb{R}^d$ denotes the region-specific transformation and bias parameters generated by the customized parameter learner.

### 4.3 Cluster-aware Masking Mechanism

Inspired by semantic-guided MAE [21], we design a cluster-aware masking mechanism to enhance the intra- and inter-cluster relation learning for our GPT-ST. The adaptive masking strategy incorporates the foregoing learned clustering information $\bar{c}_{i,r,t}$ to develop an easy-to-hard masking process. In particular, at the beginning of the training, we randomly mask a portion of regions for each cluster, in which case the masked values can be easily predicted by referring to in-cluster regions that share similar ST patterns. Subsequently, we gradually increase the masking proportion for certain categories, to increase the prediction difficulty for these clusters by reducing their relevant information. Finally, we completely mask the signals of some clusters, facilitating the ability of cross-cluster knowledge transfer for the pre-trained model. This adaptive masking process is depicted in Figure 4.

However, directly utilizing the learned cluster information $\bar{c}_{i,r,t}$ to generate the cluster-aware mask $\mathbf{M}$ is not feasible. This is because the cluster information is calculated by deeper layers of our GPT-ST network, while the generated mask is required as input to the network ($\mathbf{M} \odot \mathbf{X}$). To address this, we employ a two-layer MLP network with customized parameters to predict the learned results of $\bar{c}_{i,r,t}$. In particular, the transformations and bias vectors in this MLP network are replaced with time-dynamic and node-specific parameters generated by the customized parameter learner (Eq 4). Subsequently, a linear layer and softmax$(\cdot)$ function are used to obtain the predictions $q_{i,r,t} \in \mathbb{R}$ for $i \in H_S, r \in R, t \in T$. To optimize the distribution of $q_{i,r,t}$, we utilize the KL divergence loss function $\mathcal{L}_{kl}$ with the ground truth $\bar{c}_{i,r,t}$. It is important to note that the backpropagation of $\bar{c}_{i,r,t}$ is prevented in this step. The category is the maximum probability according to $q_{i,r,t}$ is taken as the classification result. See Algorithm 1 in A.2 for more details.

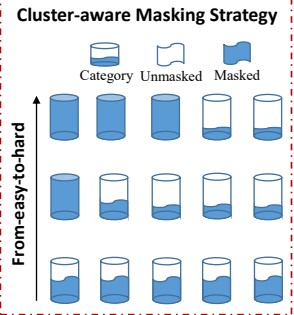

Figure 4: Illustration for the adaptive masking strategy.

## 5 Evaluation

In this section, we verify the effectiveness of GPT-ST by answering the following five questions:

- **RQ1**: How does our proposed pre-training framework GPT-ST benefit different downstream baseline methods, in terms of prediction accuracy on diverse spatio-temporal prediction tasks?

- **RQ2**: How effective are the designed technical modules of our GPT-ST framework?

- **RQ3**: How are the interpretation ability of the learned global region clusters and the inter-cluster transition patterns in the hierarchical spatial pattern encoding of our GPT-ST framework.

- **RQ4**: How efficient is GPT-ST in both the pre-training stage and the downstream task stage?

- **RQ5**: How does varying the mask ratio setting affect the performance of GPT-ST?

## 5.1 Experimental Setting

To evaluate the effectiveness of our proposed method, we conduct experiments on four real-world ST datasets including: **PEMS08** [32], **METR-LA** [23], **NYC Taxi** and **NYC Citi Bike** [46], which records the traffic flow, traffic speed, taxi order records, and bike order records, respectively.

We choose Mean Absolute Error (MAE), Root Mean Square Error (RMSE) and Mean Absolute Percentage Error (MAPE) as the evaluation metrics, which are widely used in ST prediction tasks [52]. The lower value of the metrics, the better performance of the model. Following previous works [13, 32, 22], the number of time slots $L$ is set to 12. We divide METR-LA dataset into training set, validation set and test set in a ratio of 7:1:2 and 6:2:2 for others datasets. The final model parameters are selected according to the optimal effect (minimum MAE value) of the validation set.

For optimization, in the pre-trainning phase, the absolute error loss function $\mathcal{L}_r$ is used to optimize the parameters and a hyperparameter $\lambda$ is adopted to balance the weight of $\mathcal{L}_r$ and $\mathcal{L}_{kl}$. In the downstream task stage, we firstly fuse the embedding $\zeta$ and the raw signal $\mathbf{X}$ as the input to the downstream model, and the optimization strategies are distinct for different downstream models. Detailed descriptions of datasets and evaluation are presented in the supplementary material A.1.

## 5.2 Main Results (RQ1)

In this section, we mainly investigate whether GPT-ST improves the forecasting for downstream tasks. To achieve this, we evaluate both the original performance and enhanced performance (w/ GPT-ST) of different baselines on four datasets, and the results are shown in Table 1. Note that since DMVSTGCN, STMGCN and CCRNN are specially designed for demand forecasting, we eliminate them in traffic forecasting tasks. The results indicate that the proposed model significantly improves the prediction performance of different downstream baselines on all datasets, which confirms the effectiveness of the our framework. We analyze the promotion effect of GPT-ST from three dimensions:

**Consistent improvements by GPT-ST**. We observe that GPT-ST improves different types of methods (*e.g.*, GNN-based, or attention-based models), and such positive effect does not partialize to a certain category of baselines, which verifies the generalization ability of GPT-ST. We attribute this improvement to the MAE pre-training with the awareness of intra- and inter-cluster ST relations.

**Difference among baseline models.** When compared to recently proposed methods like MSDR, our GPT-ST exhibits more substantial improvements when applied to classical baselines such as STGCN. Specifically, in traffic flow prediction using the PEMS08 dataset, MSDR shows improvements of 0.55 in MAE, 0.60 in RMSE, and 0.21% in MPAE, while STGCN demonstrates improvements of 1.61 in MAE, 2.71 in RMSE, and 0.63% in MPAE. One possible explanation for these findings is that advanced baselines like MSDR are already well-designed and comprehensive in modeling various factors. Consequently, they are capable of encoding abundant knowledge independently, which may diminish the utility of the additional signals provided by the pre-training model. Conversely, classical baselines like STGCN may derive greater benefits from the insights offered by our GPT-ST due to their simpler design and potentially limited capacity to capture complex relationships.

**Comparison with pre-training methods.** In addition to evaluating our pre-training method against spatio-temporal prediction methods without pre-training, we also compare it with the competitive pre-training baseline STEP. This method utilizes long-term time series as input for pre-training to enhance the performance of the downstream model (GWN). Despite our GPT-ST using only short-term data during the pre-training stage, it outperforms STEP on several metrics. Furthermore, our GPT-ST demonstrates an even larger performance advantage in scenarios where long-term data is insufficient, *e.g.*, the taxi and bike demand prediction tasks. This highlights the broader applicability of our proposed GPT-ST framework, emphasizing its ability to excel in various settings.

Table 1: Overall performance comparison on different datasets in terms of *MAE*, *RMSE* and *MAPE*.

| Model / Dataset Metrics | PEMS08 | | | METR-LA | | | NYC Taxi | | | NYC Citi Bike | | |
|---|---|---|---|---|---|---|---|---|---|---|---|---|
| | MAE | RMSE | MAPE | MAE | RMSE | MAPE | MAE | RMSE | MAPE | MAE | RMSE | MAPE |
| STGCN | 17.85 | 28.64 | 11.08% | 3.55 | 7.39 | 10.02% | 5.68 | 11.55 | 38.10% | 2.04 | 3.34 | 51.10% |
| w/ GPT-ST | **16.24** | **25.93** | **10.45%** | **3.29** | **6.85** | **9.14%** | **5.09** | **10.05** | **35.21%** | **1.82** | **2.82** | **49.15%** |
| DMVSTNET | - | - | - | - | - | - | 7.73 | 14.26 | 69.86% | 2.27 | 3.65 | 61.89% |
| w/ GPT-ST | - | - | - | - | - | - | **5.27** | **9.18** | **40.56%** | **1.89** | **2.86** | **55.01%** |
| TGCN | 21.44 | 31.82 | 15.67% | 3.70 | 7.28 | 10.65% | 9.73 | 19.39 | 83.50% | 2.23 | 3.72 | 62.90% |
| w/ GPT-ST | **17.34** | **26.53** | **12.26%** | **3.16** | **6.41** | **8.88%** | **6.88** | **13.80** | **51.84%** | **2.00** | **3.22** | **58.84%** |
| ASTGCN | 17.55 | 26.97 | 11.78% | 3.57 | 7.13 | 10.19% | 6.54 | 11.81 | 52.17% | 2.06 | 3.35 | 55.18% |
| w/ GPT-ST | **15.89** | **24.87** | **10.28%** | **3.10** | **6.26** | **8.74%** | **5.90** | **10.38** | **45.78%** | **1.90** | **2.93** | **53.60%** |
| GWN | 15.29 | 24.59 | 9.98% | 3.15 | 6.44 | 8.67% | 5.64 | 9.92 | 42.19% | 1.90 | 2.98 | 53.00% |
| w/ GPT-ST | **14.72** | **24.12** | **9.42%** | **3.02** | **6.18** | **8.29%** | **5.12** | **8.89** | **37.49%** | **1.80** | **2.70** | **51.69%** |
| STMGCN | - | - | - | - | - | - | 6.53 | 12.20 | 49.38% | 1.96 | 3.12 | 55.99% |
| w/ GPT-ST | - | - | - | - | - | - | **5.18** | **9.00** | **38.38%** | **1.81** | **2.73** | **52.95%** |
| MTGNN | 15.28 | 24.70 | 9.72% | 3.12 | 6.44 | 8.60% | 5.15 | 9.19 | 36.02% | 1.84 | 2.82 | 51.45% |
| w/ GPT-ST | **14.93** | **24.46** | **9.67%** | **2.99** | **6.19** | **8.21%** | **4.90** | **8.55** | **34.97%** | **1.78** | **2.67** | **51.16%** |
| STSGCN | 17.82 | 27.88 | 11.49% | 3.58 | 7.36 | 9.95% | 6.02 | 10.98 | 46.87% | 2.04 | 3.19 | 56.73% |
| w/ GPT-ST | **16.29** | **26.14** | **10.59%** | **3.16** | **6.51** | **8.74%** | **5.00** | **8.77** | **36.28%** | **1.88** | **2.79** | **55.23%** |
| STFGNN | 17.13 | 27.32 | 11.10% | 3.29 | 6.69 | 9.14% | 5.91 | 10.64 | 46.32% | 2.03 | 3.17 | 56.51% |
| w/ GPT-ST | **15.89** | **25.78** | **10.53%** | **3.13** | **6.47** | **8.84%** | **4.97** | **8.69** | **36.37%** | **1.86** | **2.74** | **54.84%** |
| STGODE | 17.73 | 27.53 | 11.39% | 3.47 | 7.02 | 9.93% | 6.62 | 12.14 | 55.03% | 2.08 | 3.27 | 56.47% |
| w/ GPT-ST | **15.68** | **24.84** | **10.46%** | **3.19** | **6.44** | **9.30%** | **5.61** | **9.91** | **43.09%** | **1.90** | **2.89** | **52.86%** |
| CCRNN | - | - | - | - | - | - | 5.41 | 10.51 | 38.47% | 1.92 | 3.01 | 52.44% |
| w/ GPT-ST | - | - | - | - | - | - | **5.08** | **9.47** | **37.04%** | **1.82** | **2.77** | **51.33%** |
| MSDR | 16.47 | 25.63 | 10.54% | 3.19 | 6.42 | 8.83% | 5.73 | 10.45 | 42.33% | 1.93 | 2.98 | 53.44% |
| w/ GPT-ST | **15.92** | **25.03** | **10.33%** | **3.07** | **6.27** | **8.50%** | **5.27** | **9.33** | **38.90%** | **1.82** | **2.74** | **52.78%** |
| STWA | 16.16 | 25.80 | 10.55% | 3.19 | 6.56 | 8.68% | 5.25 | 9.34 | 38.08% | 1.89 | 2.96 | 50.07% |
| w/ GPT-ST | **15.29** | **24.50** | **9.81%** | **3.09** | **6.32** | **8.59%** | **5.06** | **8.82** | **36.63%** | **1.83** | **2.80** | **49.91%** |
| GWN* | 13.97 | 22.94 | 9.28% | 2.95 | 5.99 | 8.01% | 5.18 | 9.08 | 35.75% | 2.02 | 3.12 | 55.16% |
| w/ STEP | 14.08 | 23.33 | **9.07%** | 2.92 | **5.91** | 7.98% | 5.33 | 9.24 | 36.17% | 2.02 | 3.12 | 53.86% |
| w/ GPT-ST | **13.96** | **22.83** | 9.16% | **2.92** | 5.97 | **7.93%** | **5.07** | **8.82** | **34.97%** | **1.98** | **3.07** | **51.44%** |

## 5.3 Model Ablation Study (RQ2)

This section investigates the impact of major components in our GPT-ST. To evaluate the impact of different components, we re-conduct pre-training for multiple ablated variants, and evaluate the performance of the downstream method using the newly pre-trained ablated models. GWN is utilized as the downstream approach, and two datasets METR-LA and NYC Taxi are employed for evaluation. The results are shown in Figure 5. From the results, we make the following observations:

**Impact of basic components.** i) -P. We remove the customized parameter learner in GPT-ST. ii) -C. We disable the hypergraph capsule clustering network to cancel the clustering operation. In this case, the spatial hypergraph NNs works on fine-grained regions directly. iii) -T. We leave the cross-cluster relation learning unexplored by removing the high-level spatial hypergraph. The noticeable performance decay demonstrates the positive benefits brought by all three components, indicating that generating personalized ST parameters and modeling the intra- and inter-cluster ST dependencies can effectively capture complex ST correlations to benefit the predictions. Among the three components, the removal of the hypergraph capsule clustering network causes the most significant performance drop. This is because the clustering results also play an important role in many other components, including the cross-cluster dependencies and the cluster-aware adaptive masking.

**Impact of mask mechanism.** i) Ran0.25 and Ran0.75. We replace our adaptive mask strategy with random masking with a mask ratio of 0.25 (mask ratio same as ours) and 0.75 (mask ratio used in MAE [15] and STEP [31]), respectively. The results consistently demonstrate that our proposed mask strategy outperforms the random mask strategies. This can be attributed to the fact that our mask strategy effectively promotes the learning of intra- and inter-cluster relationships by GPT-ST, resulting in the generation of high-quality representations. Similar results are obtained in additional experiments conducted on the other two datasets, as shown in Table 2. ii) GMAE and AdaMAE. We compare our approach to two variants using the mask strategies proposed by GraphMAE [16] and AdaMAE [2]. Both variants perform worse than our methods, highlighting the importance of considering spatial and temporal patterns in the masking strategy. This further confirms the superiority of our adaptive masking approach that leverages clustering information.

**Impact of pre-training strategy.** To further investigate the effectiveness of the mask-reconstruction pre-training approach in spatio-temporal pre-training, we compare it to other pre-training methods: local-global infomax and contrastive pre-training. We used DGI [35] and GraphCL [47] as baselines, which are widely recognized pre-training strategies for GNNs. By replacing GPT-ST's pre-training strategy with these methods (denoted as "r/"), we observe improvements in performance compared to

models without pre-training. This indicates the adaptability and benefits of infomax and contrastive pre-training for representation learning in our model. However, our approach, which utilizes the mask-reconstruction task, achieved the most significant performance enhancement. This can be attributed to the higher correlation between the mask-reconstruction task and the downstream regression task, leading to more effective learning of spatio-temporal representations. Additionally, our adaptive mask strategy plays a crucial role in facilitating the model to learn robust spatio-temporal representations by increasing the difficulty of the pre-training task.

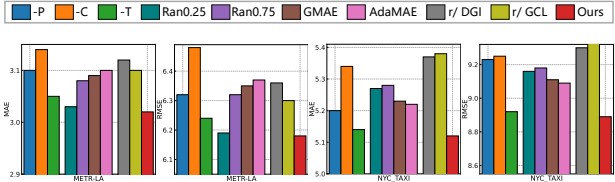 Table 2: Comparison of Ran & Ada mask.

Figure 5: Ablation study of GPT-ST.

| Dataset | PeMS08(MAE/ RMSE) | | |
|---|---|---|---|
| model | GWN | MSDR | STWA |
| Ran mask | 14.80/ 24.34 | 16.50/ 25.63 | 15.39/ 24.85 |
| Ada mask | **14.72/ 24.12** | **15.92/ 25.03** | **15.29/ 24.50** |
| Dataset | NYC Citi Bike(MAE/ RMSE) | | |
| model | DMVSTNET | STMGCN | CCRNN |
| Ran mask | 1.92/ 2.89 | 1.82/ 2.74 | 1.85/ 2.86 |
| Ada mask | **1.89/ 2.86** | **1.81/ 2.73** | **1.82/ 2.77** |

## 5.4 Investigation on Clustering Effect (RQ3)

To comprehensively demonstrate the effectiveness of the clustering process, we assess the interpretability of our GPT-ST framework by analyzing the embeddings generated by the Hypergraph Capsule Clustering Network (HCCN), as illustrated in Figure 6. We employ the T-SNE algorithm to visualize the high-dimensional embeddings produced by HCCN, mapping them to 2-dimensional vectors. Each category is represented by a distinct color, with 10 categories defined for hyperparameter $H_S$. The region clustering is determined based on the probabilities of belonging to different categories. Upon examining the visualized embeddings, we observe that regions belonging to the same class exhibit tight clustering within a confined space, providing empirical evidence of the robust clustering capability of our hypergraph capsule clustering network.

In another case study conducted on the METR-LA dataset, we investigate the in-cluster region relations derived from the hypergraph capsule clustering network, as well as the cross-class dependencies obtained from the cross-cluster hypergraph network. The results, illustrated in Figure 7, demonstrate that the top-3 regions within the same category, as depicted in Figure 7(a) and 7(b), exhibit similar traffic patterns, indicating shared functionalities. For instance, regions near commercial areas (7(a)) experience an evening peak, while those near residential areas (7(b)) remain relatively stable. These observations align with real-world scenarios. Furthermore, focusing on Figure 7(c), we analyze the top-2 regions from two categories that have undergone shifts in traffic patterns over a specific time period, while sharing similar hyperedge weights in the cross-class transition. The results reveal that regions undergoing pattern shifts exhibit distinct traffic patterns while maintaining close interconnections within a short driving distance. These findings provide further evidence of the cross-cluster transition learning's ability to capture semantic-level relationships between regions, reflecting real-world traffic scenarios. These advantages contribute to the generation of high-quality representations in our GPT-ST framework, leading to improved performance in downstream tasks.

## 5.5 Model Efficiency Study (RQ4)

In this section, we assess the efficiency of our model. Specifically, we measure the per-epoch training time of all methods on the PEMS08 dataset, and the results are summarized in Table 3. To ensure fairness, all experiments are conducted on a system equipped with a GTX 3090 GPU and an Intel Core i9-12900K CPU. The batch size for all methods, except those tagged with 'b8', is set to 64 (where 'b8' denotes a batch size of 8). By combining the results from Table 1, we can conclude that while most of the baselines experience a minor decrease in training efficiency when enhanced by GPT-ST, there is a significant improvement in their performance. For instance, some previous methods (*e.g*., STGCN) equipped with GPT-ST achieve comparable perfor-

Table 3: Computational time cost investigation.

| Model | time/epoch(s) | Model | time/epoch(s) |
|---|---|---|---|
| STGCN | 7.72 | GWN | 9.87 |
| w/ GPT-ST | 12.52 | w/ GPT-ST | 13.46 |
| ASTGCN | 12.47 | TGCN | 7.63 |
| w/ GPT-ST | 18.20 | w/ GPT-ST | 14.04 |
| MTGNN | 9.40 | STSGCN | 18.71 |
| w/ GPT-ST | 13.02 | w/ GPT-ST | 21.43 |
| STFGNN | 16.03 | STGODE | 22.93 |
| w/ GPT-ST | 19.54 | w/ GPT-ST | 19.45 |
| MSDR | 29.15 | STWA | 46.72 |
| w/ GPT-ST | 33.23 | w/ GPT-ST | 49.90 |
| Downstream tasks stage | | Pre-training stage | |
| GWN* b8 | 210.98 | STEP b8 | 327.78 |
| w/ STEP b8 | 425.12 | GPT-ST b8 | 40.18 |
| w/ GPT-ST b8 | 220.29 | *GPT-ST* | 12.50 |

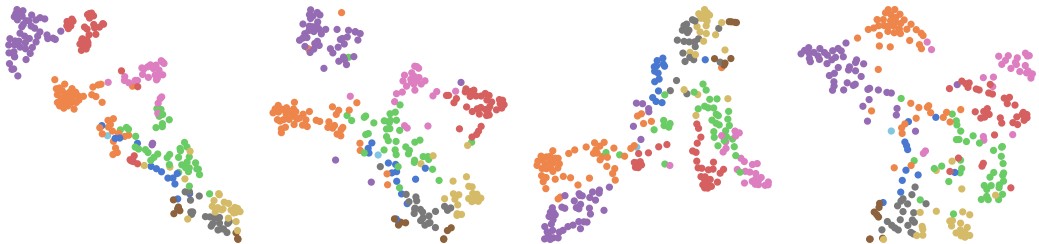

Figure 6: Visualized embbeddings of the hypergraph capsule clustering network.

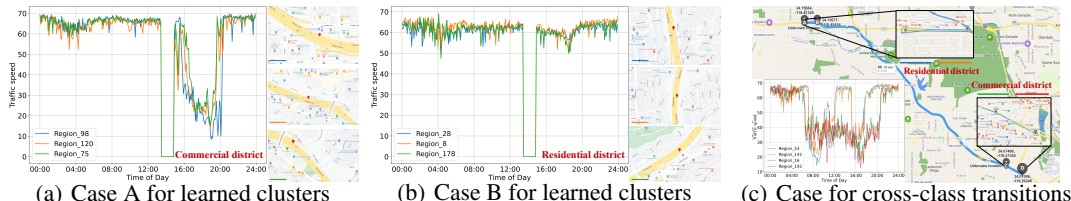

(a) Case A for learned clusters    (b) Case B for learned clusters    (c) Case for cross-class transitions

Figure 7: Case study for the spatial encoder. (a) and (b) present the real traffic conditions and their geographic locations belonging to two specific categories learned by the hypergraph capsule network; (c) illustrates the learned traffic transition relation between two categories of regions.

mance to advanced methods (*e.g.*, MSDR), while maintaining superior training efficiency. In the comparison between GPT-ST and STEP, we observe that STEP incurs a substantial time cost, both in the pre-training stage and the downstream task stage, whereas GPT-ST enhances existing baselines in a lightweight manner. These cases demonstrate that our framework achieves a win-win situation in terms of performance and training efficiency, making it well-suited for practical applications.

### 5.6 Model Hyperparameter Experiment (RQ5)

To examine the impact of different hyperparameters on GPT-ST, we conduct parameter experiments on the NYC Taxi dataset using GWN as the downstream model. The results of these experiments are depicted in Figure 8, where the evaluation metrics are plotted as relative values.

In our investigation, we specifically focus on the total mask ratio $r_t$ and adaptive mask ratio $r_a$ as the main parameters of interest. Based on the findings presented in Figure 8, we observe that the optimal effect is achieved when the total mask ratio $r_t$ is set to 0.25. Increasing the mask ratio beyond this value does not improve the model's ability to learn better representations; instead, it leads to a decline in prediction performance. Furthermore, we explore the impact of the adaptive mask ratio $r_r$, which determines the number of clusters that are entirely masked, thereby influencing the entire training process. The best performance is obtained when the adaptive mask ratio is set to 1. This suggests that this specific configuration enhances the model's ability to learn regional relations, both within individual clusters and across different clusters.

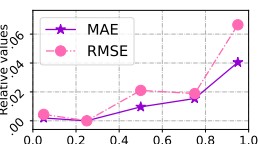

(a) Total mask ratio $m_t$

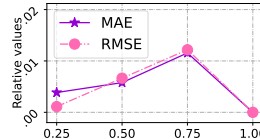

(b) Adaptive mask ratio $m_a$

Figure 8: Hyperparameter study of $m_t$ and $m_a$.

## 6 Conclusion

In this work, we introduce a scalable and effective pre-training framework tailored for spatio-temporal prediction tasks. Our framework starts with a foundational pre-training model that focuses on capturing spatio-temporal dependencies. It leverages a customized parameter learner and hierarchical hypergraph networks to extract customized spatio-temporal features and region-wise semantic associations, respectively. To further improve the model's performance, we propose an adaptive masking strategy. This strategy guides the model to learn inferential reasoning capabilities by considering both intra-class and inter-class relations during the pre-training stage. We conduct extensive experiments on four real-world datasets, demonstrating the efficacy of our proposed GPT-ST framework in enhancing the performance of downstream baselines across different spatio-temporal tasks.

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

# A  Supplementary Material

In the supplementary material, we provide additional information and details in A.1. This section covers the introduction of data, key parameter settings, comparisons with baselines, optimization methods, and the algorithm process of our method. Furthermore, A.2 presents supplementary experiments for our model, including visualization experiments and replication studies. Additionally, we discuss the reasons behind utilizing hypergraphs as the temporal encoder in A.3. Finally, the limitations and broader impacts of our work are discussed in A.4.

## A.1  Data and Implementation Details

**Data**. The statistical information of the aforementioned four real-world datasets is presented in Table 4. These datasets primarily consist of daily spatio-temporal statistics in the United States. Specifically, the **PeMS08** and **METR-LA** datasets were collected from 7 roads in the San Bernardino area and the road network of Los Angeles County, respectively. On the other hand, the **NYC Taxi** and **NYC Citi Bike** datasets were obtained from New York City.

**Parameters**. The latent representation dimensionality ($d$) and the customized parameter ($d'$) are both set to 64 and 16, respectively. The number of hyperedges ($H_T$, $H_S$, $H_M$) is set to 8, 10, and 16, respectively. We perform 2 dynamic routing iterations. Additionally, the balance ratio ($\lambda$) for $\mathcal{L}_r$ and $\mathcal{L}_{kl}$ is set to 0.1, and the total mask ratio ($r_t$) is set to 0.25.

Table 4: Statistical Information of Experimental Datasets.

| Dataset | Data Record | # Node | Time Steps | Sample Rate | Sample Date |
|---|---|---|---|---|---|
| PEMS08 | traffic flow | 170 | 17856 | 5min | 1/Jul/2016 - 31/Aug/2016 |
| METR-LA | traffic speed | 207 | 34272 | 5min | 1/Mar/2012 - 30/Jun/2012 |
| NYC Taxi | taxicab records | 266 | 4368 | 30min | 1/Apr/2016 - 30/Jun/2016 |
| NYC Citi Bike | bike orders | 250 | 4368 | 30min | 1/Apr/2016 - 30/Jun/2016 |

**Baselines**. We selected 13 methods as baselines and categorized them into distinct groups:

**Hybrid spatio-temporal prediction method:**

- **DMVSTNET** [44]: This framework for demand forecasting utilizes RNNs, convolutional networks, and fully connected layers. RNNs capture temporal correlation, convolutional networks handle spatial correlation, and fully connected layers address regional semantic correlation.

**Spatio-temporal prediction method based on GNNs:**

- **STGCN** [48]: It employs convolution operations to model both temporal and spatial dependencies.

- **GWN** [40]: This method utilizes a learnable graph structure to capture spatial dependencies and incorporates the diffuse convolution technique to capture temporal dependencies.

- **GWN\*** [40]: To evaluate the effectiveness of GWN, we utilize long-term time series data as input. Our evaluation focuses on predicting the data for the next hour using the data from the preceding two weeks. We employ a simple linear layer to process the long-term features in this prediction.

- **TGCN** [54]: In this method, both RNNs and GNNs are employed to model temporal dependence and capture spatial correlation, respectively.

- **MTGNN** [39]: This method utilizes a learnable graph structure to model associations between multiple variables. It incorporates dilated convolution and skip connections to jointly capture spatio-temporal dependencies.

- **MSDR** [25]: This model proposes a variant of RNNs to make full use of historical time step information, and combines GNNs to model long-range spatio-temporal dependence.

- **STMGCN** [11]: This method is a spatio-temporal framework designed for demand forecasting. It incorporates region association from various aspects and combines the power of RNNs and GNNs to effectively model both spatial and temporal relations.

- **CCRNN** [46]: This is an approach specifically designed for traffic demand prediction. It leverages multiple layers of GNNs, each assigned with different adjacency matrices, to capture hierarchical spatial associations. RNNs are employed to establish temporal dependencies within the network.

- **STSGCN** [32]: This work captures spatio-temporal correlations by constructing a local spatio-temporal graph, enabling the synchronous modeling of these correlations.
- **STFGNN** [22]: This work proposes a data-driven approach that utilizes a gated convolution method to generate spatio-temporal graphs. By learning spatial and temporal dependencies, the approach effectively captures the correlations within the data.

**Attention-based spatio-temporal prediction method:**

- **ASTGCN** [13]: It utilizes attention and GNNs to capture spatio-temporal periodic dependencies.
- **STWA** [4]: It integrates location-specific and time-varying parameters into the attention network to effectively capture dynamic spatio-temporal correlations.

**Spatio-temporal prediction via differential equation:**

- **STGODE** [10]: This method captures spatial dependencies by enhancing GNNs through the use of ordinary differential equations. Additionally, temporal dependencies are modeled using a dilated temporal convolutional network.

### A.1.1 Optimization Method

In the GPT-STframework, multiple temporal encoders (Section 4.1) and spatial encoders (Section 4.2) are stacked together to generate the final embeddings. The architecture includes two temporal encoders followed by a spatial encoder, forming a spatio-temporal (ST) encoding block. The final embeddings are generated by passing the data through two spatio-temporal encoding blocks.

During the pre-training phase, the predictions $\hat{\mathbf{Y}} \in \mathbb{R}^{R \times T \times F}$ are computed by applying a linear layer to the hidden dimension. To optimize the parameters, the absolute error loss function is utilized, following the approach employed in prior works such as [55, 1].

$$\mathcal{L}_r = \frac{1}{RTFr_t} \sum_{r=1}^{R} \sum_{t=1}^{T} \sum_{f=1}^{F} |(1 - \mathbf{M}_{r,t,f})(\mathbf{X}_{r,t,f} - \hat{\mathbf{Y}}_{r,t,f})| \tag{9}$$

Let $\mathbf{X}$ represent the input, consisting of $F$ features across $R$ regions in the previous $T$ time slots. The two layers of the MLP network discussed in Sec 4.3 can be formalized as follows:

$$\mathbf{Q}_{r,t} = \sigma(\sigma(\mathbf{E}_{r,t}^p \mathbf{W}_r^p + \mathbf{b}_r^p)\mathbf{W}_t^p + \mathbf{b}_t^p) \tag{10}$$

In the given formulation, $\mathbf{W}_r^p \in \mathbb{R}^{d \times d}$ and $\mathbf{b}_r^p \in \mathbb{R}^d$ represent the region-specific parameters, while $\mathbf{W}_t^p \in \mathbb{R}^{d \times d}$ and $\mathbf{b}t^p \in \mathbb{R}^d$ are the time-dynamic parameters created as in Equation 4. The predictions $q \in \mathbb{R}^{H_s \times R \times T}$ are obtained from $\mathbf{Q}$ through a linear layer followed by a softmax function. The KL divergence loss function $\mathcal{L}_{kl}$ can be formulated as follows:

$$\mathcal{L}_{kl} = \sum_{r=1}^{R} \sum_{t=1}^{T} \sum_{i=1}^{H_S} \bar{c}_{i,r,t} \cdot (\log \bar{c}_{i,r,t} - \log q_{i,r,t}) \tag{11}$$

In this case, $\bar{c} \in \mathbb{R}^{H_s \times R \times T}$ is considered as the ground truth classification result. To balance the contribution of the two loss functions, we introduce a parameter $\lambda$ to adjust their ratio, as follows:

$$\mathcal{L} = \mathcal{L}_r + \lambda \mathcal{L}_{kl} \tag{12}$$

In the downstream task stage, we utilize the pre-trained embeddings $\boldsymbol{\zeta} \in \mathbb{R}^{R \times T \times d}$ along with the raw data representation $\mathbf{E}' = \bar{\mathbf{X}} \cdot \mathbf{e}_0$ (without any mask operation) as the input for the downstream model. To fuse these two inputs, we employ a simple gated fusion layer proposed by [55], which can be formalized as follows:

$$\mathbf{H} = z \cdot \boldsymbol{\zeta} + (1 - z) \cdot \mathbf{E}; \quad z = \delta(\boldsymbol{\zeta} \mathbf{W}_{h,1} + \mathbf{E} \mathbf{W}_{h,2} + \mathbf{b}_h) \tag{13}$$

In the given formulation, $\mathbf{W}_{h,1}, \mathbf{W}_{h,2} \in \mathbb{R}^{d \times d}$ and $\mathbf{b}_h \in \mathbb{R}^d$ are learnable parameters. The variable $z$ represents the gate operation, and $\delta(\cdot)$ denotes the sigmoid activation function. It is important to note that we prevent the backpropagation of $\boldsymbol{\zeta}$ at this stage. By fusing $\boldsymbol{\zeta}$ and $\mathbf{E}'$ using the gated fusion layer, downstream models can leverage the knowledge gained during the pre-training stage for improved predictions. The specific optimization methods employed may vary depending on the downstream models, such as mean absolute error [1, 25] or Huber loss [22, 32].

### A.1.2 Algorithm Process

Algorithm 1 represents the process of the adaptive mask strategy described in Section 4.3. On the other hand, Algorithm 2 illustrates the algorithmic process during the pre-training stage.

---

**Algorithm 1:** Adaptive Mask Strategy

---

**Input:** ST data $\mathbf{X} \in \mathbb{R}^{R \times T \times F}$, maximum epoch number $E$, total mask ratio $r_t$, number of $\mathbf{X}$ elements $J$
**Output:** mask matrix $\mathbf{M} \in \mathbb{R}^{R \times T \times F}$

1 **for** $e = 1$ *to* $E$ **do**
2      Calculate the classification results $q$ from $\mathbf{X}$ according to Sec 4.3
3      Calculate adaptive mask ratio $r_a$ by $r_a = (e/E)^{\gamma}$
4      Calculate the number of masked elements by $m_t = Jr_t$, and then obtain the adaptive masked number $m_a$ and random masked number $m_r$ by $m_a = Jr_a$;   $m_r = m_t - m_a$
5      Randomly select $n$ categories from the classification list until the total number of elements of these categories is greater than $m_a$. Mask all the elements of the first $n - 1$ categories, and randomly mask the elements of the $n$-th category with the residual masked number in $m_a$
6      Randomly mask the remaining elements with random masked number $m_r$
7 **end**

---

**Algorithm 2:** Learning Process of GPT-ST Framework

---

**Input:** Spatio-temporal data $\mathbf{X} \in \mathbb{R}^{R \times T \times F}$, mask matrix $\mathbf{M} \in \mathbb{R}^{R \times T \times F}$, dynamic routing iterations number $\mathcal{R}$, maximum epoch number $E$, learning rate $\eta$
**Output:** Trained parameters in $\Theta$

1 Initialize all parameters in $\Theta$
2 **for** $e = 1$ *to* $E$ **do**
3      Calculate approximate classification results $q$ according to Eq 10 and then generate the mask matrix $\mathbf{M}$ according to Alg 1
4      Mask the elements in $\mathbf{X}$ with $\mathbf{M}$ and then calculate the initial representation $\mathbf{E}$ of the masked traffic data for each region in each time slot
5      Calculate the raw temporal features $\mathbf{d}_t$ according to Eq 4 and initialize the free-form region embedding matrices $\mathbf{c}_r$
6      Encode the temporal traffic pattern with $\Gamma$ by integrating customized parameters $\mathbf{d}_t$ and $\mathbf{c}_r$ into the temporal hypergraph neural network according to Eq 3
7      Generate the normalized region embedding $\mathbf{v}$ and calculate the transferred information $\bar{\mathbf{v}}_{i|r,t} \in \mathbb{R}^d$ from each region $r$ to each cluster center (hyperedge) $i$ according to Eq 5
8      **for** $r_n = 0$ *to* $\mathcal{R}$ **do**
9          Perform the dynamic routing algorithm to characterize the semantic similarities between the regions (low-level capsules) and the spatial cluster centroids (high-level capsules) according to Eq 6
10      **end**
11      Generate the final cluster embedding $\bar{\mathbf{s}}$ for cross-class relationships learning
12      Generate the personalized high-level hypergraph structure and conduct it on the reshaped embedding $\tilde{\mathbf{s}}$ to generate $\hat{\mathbf{s}}$ to model inter-classes relationships according to Eq 7
13      Conduct the customized low-level hypergraph structure to propagate the clustered embeddings $\hat{\mathbf{s}}$ back to the regional embeddings $\Psi$ according to Eq 8
14      Make predictions $\hat{\mathbf{Y}}$ and calculate the absolute error loss $\mathcal{L}_r$ according to Eq 9
15      Calculate the KL divergence loss $\mathcal{L}_{kl}$ based on Eq 11
16      Calculate the final loss $\mathcal{L}$ according to Eq 12
17      **for** $\theta \in \Theta$ **do**
18          $\theta = \theta - \eta \cdot \partial \mathcal{L} / \partial \theta$
19      **end**
20 **end**
21 **return** all parameters $\Theta$

---

## A.2 Additional experiments

### A.2.1 Visualization Study

Figure 9 presents the reconstruction results of the model during the pre-training stage. In the figure, gray, red, and blue lines respectively represent the masked signals, predicted values, and visible

signals. The results demonstrate that GPT-ST can accurately predict the masked signals based on the visible signals, regardless of whether a random mask or an adaptive mask is used. Furthermore, it can be observed that the masked signals generated by the adaptive strategy exhibit greater temporal consistency compared to those generated by the random strategy. This is because the category attribute of a region is less likely to change within a short period of time. The continuous mask and cluster mask are effective in increasing the difficulty of the reconstruction task, enabling GPT-ST to learn robust spatio-temporal representations even with low masking ratios.

Figure 10 summarizes the enhancement effect of your model on downstream baselines. The figure showcases the predicted performance of four baselines on the PEMS08 dataset, where the blue, green, and red lines respectively represent the ground truth, the original performance, and the enhanced performance of the baselines. With the assistance of GPT-ST, the prediction performance of the baseline models experiences significant improvements on certain subsets, as highlighted by the red box. This confirms that the pre-trained model, equipped with customized parameter learners and mechanisms for encoding intra- and inter-class spatial patterns, can provide a discriminative and semantic representation for downstream tasks. As a result, it effectively compensates for the limitations of different baselines.

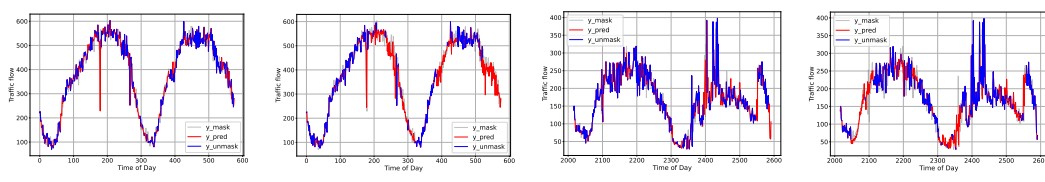

| (a) Case A for Ran mask | (b) Case A for Ada mask | (c) Case B for Ran mask | (d) Case B for Ada mask |

Figure 9: Visualization experiments of reconstruction performance in the pre-training stage.

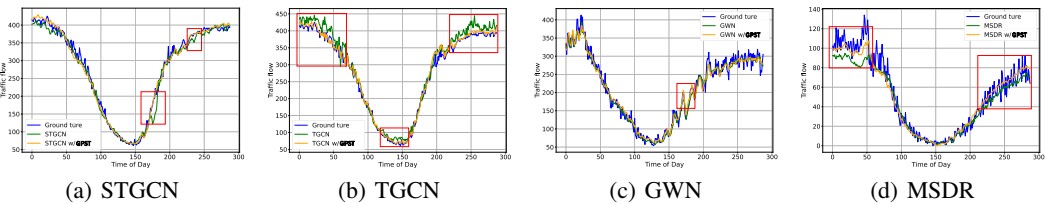

| (a) STGCN | (b) TGCN | (c) GWN | (d) MSDR |

Figure 10: Visualization experiments of different baselines in the downstream stage.

### A.2.2 Replication Study

In this section, the deviation of GPT-ST's performance over random parameter initialization is investigated. The statistical results are presented in Table 5. For each performance evaluation on the PEMS08 dataset, we randomly selected 5 seeds. The results indicate that GPT-ST exhibits strong adaptability to different parameter initialization settings, consistently providing stable and high-quality spatio-temporal representations for downstream models.

Table 5: Replication study on PEMS08 dataset.

| Model | MAE | RMSE | MAPE | Model | MAE | RMSE | MAPE |
|---|---|---|---|---|---|---|---|
| STGCN | 17.99±0.14 | 28.56±0.18 | 11.22±0.24 | GWN | 15.37±0.28 | 24.61±0.26 | 9.74±0.15 |
| w/ GPT-ST | **16.40±0.16** | **26.27±0.29** | **10.62±0.20** | w/ GPT-ST | **14.77±0.05** | **24.19±0.08** | **9.52±0.07** |
| ASTGCN | 18.05±0.30 | 27.76±0.47 | 11.40±0.26 | TGCN | 21.47±0.04 | 31.82±0.03 | 16.06±0.25 |
| w/ GPT-ST | **16.83±0.50** | **26.40±0.71** | **10.78±0.45** | w/ GPT-ST | **17.33±0.08** | **26.50±0.11** | **12.39±0.11** |
| MTGNN | 15.34±0.04 | 24.55±0.09 | 9.72±0.04 | STSGCN | 17.96±0.10 | 28.19±0.20 | 11.65±0.17 |
| w/ GPT-ST | **14.96±0.05** | **24.47±0.10** | **9.66±0.04** | w/ GPT-ST | **16.28±0.07** | **26.05±0.09** | **10.51±0.08** |
| STFGNN | 17.20±0.07 | 27.55±0.18 | 11.09±0.10 | STGODE | 17.81±0.06 | 27.64±0.08 | 11.33±0.07 |
| w/ GPT-ST | **15.90±0.03** | **25.86±0.06** | **10.42±0.16** | w/ GPT-ST | **15.89±0.16** | **25.05±0.16** | **10.59±0.32** |
| MSDR | 16.52±0.16 | 25.70±0.19 | 10.69±0.35 | STWA | 15.80±0.22 | 25.16±0.36 | 10.15±0.20 |
| w/ GPT-ST | **15.96±0.06** | **25.09±0.08** | **10.33±0.08** | w/ GPT-ST | **15.30±0.05** | **24.64±0.12** | **9.99±0.16** |

### A.3 Analysis of Hypergraph as Temporal Encoder

Existing time encoders in spatio-temporal prediction schemes, such as recurrent neural networks (RNNs), temporal convolutional networks (TCNs), and attention mechanisms, have certain limitations. RNNs are prone to losing long-term information due to the vanishing gradient problem. TCNs can only capture temporal information within a limited neighborhood defined by the size of the convolution kernel. Although attention mechanisms consider the interrelationship between time steps, their computational efficiency decreases significantly when the length of the time series ($T$) is large, resulting in a time complexity of $O(T^2)$. These limitations hinder the ability of existing approaches to effectively capture and model the temporal dynamics in spatio-temporal data.

To address the aforementioned limitations, we propose to utilize the hypergraph neural network as a temporal encoder for time-dependent modeling. A hypergraph is composed of multiple sets of hyperedges, where each hyperedge serves as a learnable information hub connecting time-step information with different weights. By treating different hyperedges as a means of integrating temporal relationships across different dimensions, the hypergraph neural network offers a flexible control over complexity by adjusting the number of hyperedges. This approach allows for modeling long-term temporal dependencies while maintaining low computational costs. The hypergraph neural network combines the advantages of capturing temporal dynamics effectively and efficiently, making it a promising solution for spatio-temporal prediction tasks.

### A.4 Limitations and Broader Impacts

The proposed GPT-ST has demonstrated its effectiveness in improving the prediction performance of downstream models. However, it also has two main limitations: i) Task-specific pre-training: GPT-ST requires pre-training for each specific downstream task. This is because different prediction tasks often have distinct data formats and distributions, necessitating task-specific pre-training. For instance, a GPT-ST pre-trained on task A cannot be directly applied to task B. ii) Increased time cost: Both the pre-training process and the enhancement process in the downstream task stage add to the prediction time cost. These additional computations can impact real-time applications or scenarios with stringent time constraints. In future research, we aim to explore more generalized and versatile spatio-temporal pre-training frameworks, along with lightweight algorithms, to address the task-specific pre-training requirement and further reduce the computational overhead.

