# OpenReview forum: "GPT-ST: Generative Pre-Training of Spatio-Temporal Graph Neural Networks"
_NeurIPS.cc/2023/Conference — NeurIPS 2023 poster_

### Official Review · Reviewer_83m7 · 2023-07-04

**Soundness:** 3 good
**Presentation:** 3 good
**Contribution:** 3 good
**Rating:** 7
**Confidence:** 4

**Summary:**

This paper proposes a novel pre-training framework (GPST) for ST prediction, it mainly integrates the ST parameter personalization scheme and the region-wise semantic association mechanism into the pre-training model, and conduct the training in an unsupervised manner. Besides, the proposed model GPST adopts the hierarchical hypergraph structure to capture the semantic-level association of multiple regions from a global perspective. Extensive experiments are conducted.

**Strengths:**

1. This paper is well-presented and well-organized.
2. This paper introduces a spatio-temporal pre-training framework that can be easily integrated into downstream baselines and improve their performance.
3. Detailed experimental results are provided to validate the model.


**Weaknesses:**

1. The model's performance seems to heavily rely on the pre-training stage. If the pre-training is not done correctly or if the pre-training data is not representative of the test data, the model's performance could suffer, how can this case be handled?
2. Why not include POIs, which should benefit the prediction performance.

**Questions:**

See above.

**Limitations:**

Noone

---

> ### Author Rebuttal · Authors · 2023-08-10
>
> ***Response to Reviewer 83m7***
> **Comment 1:** Further discussion on the design of the pre-training stage and data distribution.
> **Response:** Thank you for your feedback. This work primarily introduces a spatio-temporal pre-training framework to enhance downstream baselines, where the effectiveness of the enhancement is highly dependent on the designs of the pre-training stage. If the pre-training model or mechanisms are not appropriately designed, the embeddings generated by the model may lack superior representational capacity, consequently compromising the performance of downstream baselines. To address this issue, we address the limitations of prior research in spatio-temporal modeling and comprehensively consider these concerns in the construction of the pre-training model. In terms of the pre-training mechanism,  an adaptive mask strategy is proposed to guide the model to learn robust spatio-temporal representations.
> The performance of most models suffers significantly if the training data fails to represent the test data, which remains a major challenge in the field of machine learning. Following prior works (such as GWN[1] and STEP[2] ), we assume that the training (or pre-training) and test data distributions are approximately similar, and this work does not address the scenario where there exists a significant distribution discrepancy between the training and test data. There exists a research line dedicated to exploring this issue (such as domain adaptation, domain generalization, and out-of-distribution (OOD)), which is  an intriguing research direction. In future work, we will continue to investigate the related issues concerning the generalization ability of spatiotemporal models (such as different distributions between test and training data, data distribution drift, and so on).
>
> [1] Graph wavenet for deep spatial-temporal graph modeling. IJCAI 2019.
> [2] Pre-training enhanced spatial-temporal graph neural network for multivariate time series forecasting, KDD 2022.
>
> **Comment 2:** Explanation for the exclusion of POIs.
> **Response:** Thank you for your constructive comment. Intuitively, incorporating POIs information can enhance the predictive performance of the model. However, we did not utilize such information primarily based on the following two considerations. i) Difficulties in obtaining POIs data. Due to the absence of POIs in the utilized dataset, their inclusion would require manual collection. However, collecting and determining POIs data in real-world urban scenarios can be extremely challenging. This is primarily attributed to the dataset's extensive coverage of numerous urban regions with significant geographical spans, demanding substantial human resources and time commitment to accomplish this task. ii) Fairness of experiments. Previous works did not incorporate POIs when evaluating on the utilized datasets. To present a fair comparison and showcase the improvement achieved by our model, we refrained from introducing additional artificially created data during the pre-training stage.

---

> > ### Comment · Reviewer_83m7 · 2023-08-11
> > **Thanks for rebuttal**
> >
> > The responses from the authors have clearly addressed the problems pointed out previously. Thanks for all responses.

---

> > > ### Author Response · Authors · 2023-08-11
> > >
> > > Thank you very much. If you have any further questions regarding this work, please feel free to engage in discussions with us.

---

### Official Review · Reviewer_8n4n · 2023-07-06

**Soundness:** 2 fair
**Presentation:** 1 poor
**Contribution:** 2 fair
**Rating:** 4
**Confidence:** 1

**Summary:**

This paper introduce pre-training task for spatio-temporal. The algorithm design includes a spatial-temporal knowledge extractor and a pre-training mechanism.

The design of the pre-training methods is to resolve the two limitations of existing spatial-temporal learning: (1) lack of comprehensive personalization (2) insufficient consideration of modelling semantic.



**Strengths:**

This paper study an important problem: GNN-pretraining.

**Weaknesses:**

1. The presentation wasn't very clear, please refer to questions.

2. Missing many recent baselines, e.g., TGAT, JODIE, TGN. The authors can compare with those works using https://github.com/amazon-science/tgl. Besides, there are most more recent methods [1-3] need to be included in discussion.

[1] Inductive Representation Learning in Temporal Networks via Causal Anonymous Walks
[2] Provably expressive temporal graph networks
[3] Do We Really Need Complicated Model Architectures For Temporal Networks?

3. Experiment dataset size are all relatively small. I am not sure this method can scale to very large graph from experiments.



**Questions:**

1. Could you please explicitly summarize what is the SSL pre-training method different from the method on static graph? My impression is the main contribution is due to spatial-temporal vs static graph setting. The mask and predict

2. The hyper-edge definition is not very clear to me. At line 116 "each of which connects multiple vertices", could you please explain how an edges can connect to more than 2 nodes?

3. How intra-class pattern and inter-class relation are defined (line 138). Is this coming from dataset, or the authors define them based on human knowledge per dataset.

4. Is the pre-training task a link prediction task? Do you explicitly consider the potential information leakage issue during pre-training. For example, the positive edges for pre-training might be sampled as a neighbor when computing the node representation.

5. Will GNN size affect the pre-training efficiency?

---

> ### Author Rebuttal · Authors · 2023-08-10
>
> ***Response to Reviewer 8n4n***
> **Comment 1:** The difference between our pre-training method and the static graph approach.
> **Response:** Thank you for your suggestion. Compared to static graphs, spatio-temporal (ST) graphs focus more on uncovering the temporal evolution patterns of node features and the dynamic correlations between nodes. The differences between these two approaches primarily lie in the design of the models and the pre-training strategies.
> (1) Model design. Static graph models typically rely on time-agnostic GNNs for capturing fixed graph structures without time information. However, in ST graph models, it becomes crucial to capture node-specific temporal information and time-evolving spatial dependencies. To tackle these challenges, our GPST employs a temporal encoder and personalized parameter learners to capture the dynamics of node features. Additionally, GPST utilizes a hierarchical hypergraph network to capture the evolving patterns of region-wise spatial relations. As a result, GPST can effectively encode complex ST relations that are not adequately addressed by static graph models.
>
> (2) Pre-training strategy: Pre-training in static graph methods typically involves masked autoencoding for features and structures. However, in ST modeling, handcrafted structure information is often noisy and incomplete. This limits the effectiveness of structure reconstruction for pretraining in ST graphs. To address this, we propose an adaptive mask strategy based on ST data for feature autoencoding. This strategy enables the reconstruction of valuable knowledge in an easy-to-hard manner, facilitating high-quality ST pre-training.
>
> **Comment 2:** Further explanation of hyperedge.
> **Response:** Thank you for your feedback. In contrast to conventional edges that transfer information between two nodes, hyperedges expand the notion of edges to act as intermediate information hubs among multiple nodes [1-2]. Specifically, the hyperedge connections is denoted $\textbf{H}\in\mathbb{R}^{N\times H}$, where $N$ and $H$ are the number of nodes and the number of hyperedges. Each hyperedge captures the connectivity among nodes and hyperedges, with messages initially propagated from all nodes to each hyperedge, followed by transmitting aggregated messages back to each node.
>
> [1] Hypergraph Neural Networks
> [2] Dynamic Hypergraph Neural Networks
>
> **Comment 3:** Further explanation of intra-class pattern and inter-class relation.
> **Response:** Thank you for your comment. Relevant concepts are mentioned in the introduction (starting from line 44). In urban settings, similar regions exhibit comparable ST patterns (e.g., residential areas sharing similar traffic patterns), while there are also some associations between different categories of regions (e.g., people moving from residential areas to office areas). These patterns are neither directly obtained from the dataset nor artificially defined. It is a potential knowledge that we aim to extract during the training process, achieved by the iterative clustering effect of capsule networks [3] integrated into the hypergraph network.
>
> [3] Dynamic routing between capsules
>
> **Comment 4:** Further clarification of the pre-training task.
> **Response:** Thank you for your feedback. The pre-training task is to mask and reconstruct node features, which is meticulously defined and explained in Preliminaries (Section 3 starting from line 121).  Upon thorough verification, we have confirmed the absence of information leakage in this work. Additionally, since our pre-training task does not involve link prediction, the mentioned sampling-induced information leakage scenario does not occur.
>
> **Comment 5:** Further discussion on the impact of GNN Size on the proposed model.
> **Response:** Thank you for your suggestion. The complexity of our Intra- & Inter-class spatial patterns encoding module (spatial encoder) is proportional to the number of time slots $T$ and the number of regions $R$, similar to a vanilla GCN. Hence, our model exhibits scalability to large datasets, akin to other graph architectures.
>
> **Comment 6:** Clarification of recent methods for temporal graph networks.
> **Response:** Thank you for your suggestion. We observe that the mentioned methods predominantly involve works related to temporal graph networks (TGNs), rather than the field of ST prediction that we are concerned about, and we will elaborate from the following two perspectives.
> (1) Why we did not consider the mentioned methods as baselines and include them in the discussion of this paper?
> i) There is a gap between the tasks addressed by TGNs and ST prediction. ST prediction primarily deals with forecasting future values based on historical data in a spatial and temporal context. While temporal graph networks focus on dynamic reasoning over graph structures (e.g. link prediction). Different tasks require tailored model architectures to meet their specific requirements.
> ii) We followed the experimental setup used in existing research on ST prediction [4-5], and the baseline designed specifically for ST prediction tasks may be more representative.
>
> [4] Spatial-temporal fusion graph neural networks for traffic flow forecasting
> [5] MSDR: Multi-step dependency relation networks for spatial temporal forecasting
>
> (2) We adapted the mentioned baselines to suit the ST prediction task and discussed the works related to TGNs.
> i) Additional comparative experiment of JODIE and TGN. Results are presented in the following table (in terms of MAE).
>
> | model/dataset | PEMS08 | METR-LA | NYC TAXI | NYC Citi Bike |
> |:--:|:--:|:--:|:--:|:--:|
> | JODIE | 18.53 | 3.45 | 6.92 | 2.09 |
> | w/ GPST | 16.59 | 3.11 | 6.19 | 1.90 |
> | TGN | 19.33 | 3.61 | 7.58 | 2.09 |
> | w/ GPST | 16.11 | 3.16 | 5.94 | 1.89 |
>
> ii) Related works. Thank you for your suggestion. Due to character limitations, we include the discussion of related works in the global response.

---

> > ### Comment · Reviewer_8n4n · 2023-08-10
> >
> > Q1: I think I didn’t make myself clear in the previous feedback. Let me rephrase in another way. Let’s say the time-encoding in Eq. 3 is a contribution as mentioned by author. But if we look at Eq. 3, its just applying an MLP on the input node features, and here the input node features is named as day feature $z^{(d)}$ and week feature $z^{(w)}$. By replacing the $z^{(d)}, z^{(w)}$ with any node features $x$, it looks exactly a static graph algorithm. The same for adaptive mask strategy, I am use it exactly for static graph.
> >
> > Q2: I am still confused. You mean hyper-edge is actually like another node. For example if hyper-edge connect node 1 with node 2+3, then its like (1, hyper-edge), (hyper-edge, 2), (hyper-edge, 3)?
> >
> > Q3: Sorry, I know what is intra-class pattern and inter-class relation. My original question is “Is the intra-class pattern and inter-class relation coming from dataset, or the authors define them based on human knowledge per dataset.” How do you know this exists? Any evidence?
> >
> > Q4: I was asking this because most graph learning model use link prediction to explicitly learn the graph structure. This is often the case for most temporal graph network methods: pre-train on link-prediction then use it for node-classification.
> >
> > Q5: Most pre-training method works well on large model, e.g., the ResNet in vision task and Bert Models in language task, and it doesn’t often works well on GNN. For example if we take a look at open graph benchmark, none of those methods are using pre-training. One of the hypothesis is GNN is small, not suitable for pre-training. Therefore, many graph transformers are proposed and they show pre-training on those large GNN models indeed help. My question originally want to ask whether you have same observation. I should use “effectiveness” instead of “efficiency”.
> >
> > Q6: Temporal graph networks can also handle node classification task if I recall correctly. Could you please elaborate how GPST are integrated with JODIE and TGN?

---

> > > ### Author Response · Authors · 2023-08-11
> > >
> > > Thank you very much for your response. We will address the issues you have raised.
> > >
> > > **Response to Q1:**  Thank you for your feedback. The graph structure varies across different time steps in our method, distinguishing it from static graphs that remain unchanged over time.  In specific, Eq (3) demonstrates the generation of temporal features based on actual time information. These features can be integrated with the hypergraph structure to produce the dynamic hypergraph structure. For example, the hypergraph connectivity $\textbf{H}'_t$ is customized to the $t$-th time slot using another initial temporal embeddings $\textbf{D}_t^{'}$ by $\textbf{H}'_t=\text{softmax}(\textbf{D}_t^{'\top} \cdot \bar{\textbf{H}}')$, where $\bar{\textbf{H}}'\in\mathbb{R}^{d'\times H_S\times R}$ contains embeddings for each hypergraph connection (Section 4.3.1 start from line 205). Here, $\textbf{H}'_t\in\mathbb{R}^{H_S\times R}$ (also $\textbf{H}'\in\mathbb{R}^{T \times H_S\times R}$) represents the generated dynamic hypergraph structure. This structure is employed for personalized message aggregation on node features at different time steps. However, for static graphs, the aforementioned process does not exist. The proposed adaptive mask strategy exhibits spatio-temporal awareness as it is based on the aforementioned dynamic hypergraph structure.
> > >
> > > **Response to Q2:** Thank you for your response. The concept of hyperedges and the mentioned another type of node share some similarities. It is akin to a virtual node at another hierarchical level that associates multiple actual nodes. The dinsinction lies in that hyperedges do not represent concrete entities (e.g. regions, roads, intersections) in the physical world, but nodes are usually used to represent entities in the observed data. Furthermore, related works [1-2] have indicated the effectiveness of hypergraphs in clustering or latent factor mining.
> > >
> > > [1] Inhomogeneous Hypergraph Clustering with Applications, NIPS 2017
> > > [2] Hypergraph Clustering Based on PageRank, KDD 2020
> > >
> > > **Response to Q3:** Thank you for your comment. In brief, these patterns are not labeled in the dataset. Instead, we expect the model to learn these latent relations. Our case study serves as evidence of GPST's capability to discover such patterns.
> > > Specifically, the employed dataset records urban ST data (e.g. traffic flows). Similar regions within the city exhibit analogous traffic patterns (e.g., high traffic during holidays in commercial districts), while dissimilar regions show interdependencies (e.g., people moving from residential to office areas during peak hours), representing normal human activities. Such patterns are unlabeled in the dataset. Thus, we empower GPST with the capacity to capture these patterns by constructing a learnable hierarchical hypergraph network. We confirm this through the case study (Section 5.3). In Figures 5(a) and 5(b), GPST groups three regions with akin traffic patterns into a single category. The latitude and longitude data further affirm their common classification, like being commercial districts. And Figure 5(c) illustrates potential traffic migration between different-category regions captured by GPST.
> > >
> > > **Response to Q4:** Thank you for your response. If you have any questions about pre-training tasks, please feel free to discuss them with us.
> > >
> > > **Response to Q5:** Thank you for your valuable feedback. We fully recognize the significance of having sufficient and high-quality training data for achieving successful pre-training. In the domain of ST prediction tasks, although the dataset may not contain a large number of nodes, each individual node encompasses a substantial amount of temporal features. For instance, the METR-LA dataset records traffic data over a four-month period with a sampling frequency of 5 minutes. This rich historical data serves as a solid basis for the pre-training process of GPST, and its ability to effectively enhance downstream baselines further substantiates this observation.
> > >
> > > **Response to Q6:** Thank you for your reply. Similar to other baselines, we utilize the representations generated by GPST as inputs to enhance the performance of JODIE and TGN (appendix A.1.3). The implementation details for JODIE and TGN are as follows.
> > > (1) JODIE: We assigned static embeddings to each node (or region) and used RNNs to handle the dynamic embedding updates of node features, where GNNs replaced the interactions between users and items. A time encoding layer combined with embedding mapping operations was used to update the dynamic embeddings of nodes to accommodate predictions at different time steps.
> > > (2) TGN: MLPs and RNNs were employed as the message and memory functions for updating node features. A multi-head attention mechanism was utilized to aggregate messages from neighbors (constructed based on distance).
> > >
> > > Thank you again. We welcome further discussion and are available to address any questions you may have.

---

> > > > ### Comment · Reviewer_8n4n · 2023-08-15
> > > > **Thank you for the responses.**
> > > >
> > > >  The authors tried their best to answer all my questions, and most of my concerns are addressed.But I have to say that not all answers given by authors make sense to me (e.g., major contribution mentioned in Q1, evidence intra- and inter-class pattern is important instead of only intuitions in Q3, why only consider mask-and-prediction but not other graph pre-trianing in Q4, impact on model size in Q5), but this could potentially because I am not familiar with the spatial-temporal setting used in this paper, and may not correctly evaluated this paper as other reviewers.
> > > >
> > > > This paper is more like a data-mining paper that apply existing ML algorithms on a spefic real world problem to solved it: This paper take mask-and-predict to solve a traffic prediction problem and using some dataset spefic observation to get better performance.
> > > > I have to say it is not very easy to digest this paper, and I believe it belongs to data mining conferences (e.g., ICDM, WWW, KDD) more than ML conferences. I was expecting to papers working on **theory** or **empirical study** to reveal some more **general ML problems** (e.g., consider spatial-temporal as a more general setting instead on solving a specific downstream task, e.g., traffic prediction). Therefore, I increased my score slightly given the effort authors made during rebuttal, and I will also reduced my confidence score because I am not familiar with the specific setting in this paper.
> > > >
> > > > Suggestions:
> > > > In the revised version, the authors might want to try other popular pre-training methods, espetially those designed for MPNN with an ablation study to justify why mask-then-prediction is a good choice.

---

> > > > > ### Author Response · Authors · 2023-08-18
> > > > > **Response to Reviewer 8n4n**
> > > > >
> > > > > We sincerely appreciate your positive feedback. The improvement of the score is a profound affirmation and encouragement for us, and your suggestions are exceptionally insightful. We are fully committed to addressing any questions or concerns you may have regarding this work.
> > > > >
> > > > > ***Comment 1: Discussion of the mentioned questions.***
> > > > > **Response:** We appreciate your response. Regarding Q3, we consider the intra- and inter-class patterns to be manifestations of the regularities exhibited by urban activities. (e.g., regions with similar functionalities exhibit similar spatio-temporal patterns). Such regularity can be derived from empirical observations of human activities. We will address Q4 in Comment 3. If there are any specific questions or concerns regarding Q1 and Q5, we will make every effort to address them.
> > > > >
> > > > > ***Comment 2: Generalizability of GPST.***
> > > > > **Response:** We sincerely appreciate your feedback. In this paper, we propose a spatio-temporal pre-training framework to enhance the performance of downstream spatio-temporal models. Following the settings of previous spatio-temporal graph studies [1-2], we focus on the specific task of spatio-temporal prediction in real urban scenarios. Real-world data distributed on spatio-temporal dimensions (e.g. traffic, crime, house pricing) has their unique but common patterns. For example, spatio-temporal features exhibit diverse evolutionary patterns across different time periods, and regions with similar functionalities tend to demonstrate similar spatio-temporal patterns. This paper aims at excavating such patterns with a unified pre-training framework. And our empirical study suggests its generalization ability across a variety of different tasks.
> > > > >
> > > > > [1] Graph wavenet for deep spatial-temporal graph modeling, IJCAI 2019.
> > > > > [2] MSDR: Multi-step dependency relation networks for spatial temporal forecasting, KDD 2022.
> > > > >
> > > > > ***Comment 3 Additional Ablation Experiments.***
> > > > > **Response:** Thank you for your suggestion. In the Model Ablation Study (Section 5.3), we compared our approach with several generative pre-training strategies, such as GraphMAE. To further investigate why mask-reconstruction is a good choice, we extended our exploration to other types of pre-training approaches, specifically contrastive pre-training. Contrastive learning is a widely adopted pretraining strategy for representation learning with the techniques of data augmentation and information maximization. We selected DGI[3] and GraphCL[4] as baselines, which are two widely recognized and extensively used pretraining strategies in GNNs.
> > > > > DGI: This method learns node representations in graph-structured data by maximizing the mutual information between local representations and global summaries.
> > > > > GraphCL: It incorporates four data augmentation techniques to generate two contrasting views, optimizing the model by maximizing the mutual information between the views.
> > > > >
> > > > > We conducted ablation study on the METR-LA and NYC TAXI datasets, and the results are presented in the table below. Here, "r/" denotes the replacement of the GPST pretraining strategy with the DGI or GraphCL pretraining strategy. The results demonstrate the effectiveness of contrastive pre-training methods in enhancing downstream models, highlighting the adaptability of contrastive learning strategies in our model for representation learning. Notably, our approach achieves the most significant performance enhancement, possibly attributed to the higher correlation between the mask-reconstruction task and the downstream task (regression task). In addition, the proposed adaptive mask strategy also contributes to the model to learn robust spatio-temporal representations.
> > > > >
> > > > > Results on METR-LA dataset
> > > > > | Model/Metrics | MAE | RMSE | MAPE |
> > > > > |:--:|:--:|:--:|:--:|
> > > > > | GWN | 3.15 | 6.44 | 8.67% |
> > > > > | r/ DGI | 3.12 | 6.36 | 8.64% |
> > > > > | r/ GraphCL | 3.10 | 6.30 | 8.50% |
> > > > > | w/ GPST | **3.02** | **6.18** | **8.29%** |
> > > > >
> > > > > Results on NYC TAXI dataset
> > > > >
> > > > > | Model/Metrics | MAE | RMSE | MAPE |
> > > > > |:--:|:--:|:--:|:--:|
> > > > > | GWN | 5.64 | 9.92 | 42.19% |
> > > > > | r/ DGI | 5.37 | 9.30 | 40.20% |
> > > > > | r/ GraphCL | 5.38 | 9.55 | 38.49% |
> > > > > | w/ GPST | **5.12** | **8.89** | **37.49%** |
> > > > >
> > > > > [3] Deep Graph Infomax, ICLR 2019.
> > > > > [4] Graph Contrastive Learning with Augmentations, NIPS 2020.

---

### Official Review · Reviewer_fFpq · 2023-07-06

**Soundness:** 3 good
**Presentation:** 2 fair
**Contribution:** 2 fair
**Rating:** 6
**Confidence:** 4

**Summary:**

The authors introduce a spatio-temporal pre-training framework that can be easily integrated into downstream baselines. The framework comprises personalized parameter learners and hierarchical hypergraph networks. The former enables the acquisition of spatio-temporal personalized representation and the latter capture the intra-class and inter-class correlations among regions. Besides, the authors design an adaptive mask strategy to guide the model in learning diverse relationships across regions. Experiments are conducted on representative benchmarks on four real-world datasets to evaluate the proposed framework.

**Strengths:**

1. The authors propose a novel pre-training framework for Spatio-Temporal prediction which can be easily applied to existing advanced ST neural networks.

2. The authors devise an adaptive mask strategy that guides the model in a progressive manner, facilitating the acquisition of useful knowledge from other categories to recover unknown categories. The comparisons with other mask strategies demonstrate the superiority of the designed mechanism.

3. The authors evaluate both the original performance and enhanced performance with the proposed framework of different baselines on four datasets. The results indicate that the proposed model significantly improves the downstream baseline prediction performance and confirm the effectiveness of the framework.


**Weaknesses:**

1. The presentation of this paper should be improved:

    (1) The explanations of some concepts are not clear. For instance, the meaning of the symbol $M_{r,t}$ in Section 4.2 is unclear. Does it represent the mask operation? Furthermore, the first-level hyperedges mentioned in Section 4.3.2 lack clear explanation, as it is the first occurrence of the term "first level" in the paper.

    (2) Equation (4) is referenced multiple times in the paper to illustrate the process of generating embeddings. However, it is necessary to provide a more detailed explanation of how the equation is actually implemented. For instance, in Line 173 of Section 4.2, it would be helpful to provide specific equations for the computations involved. The inclusion of concrete examples would greatly enhance the understanding of Equation (4) and its functionality.

    (3) In Section 4.2, there are multiple subscripts used, including (r,t), r, and t. It would be better if provide a more detailed explanation to clarify the differences between these descriptions.

    (4) In Section 5.1.1, the paper mentions the use of the absolute error loss function to optimize the parameters. However, there is a lack of detailed description regarding how this loss function is precisely employed. It would be beneficial to provide a more thorough explanation of how the optimization process utilizes the absolute error loss function, including any specific computations.

2. The authors adopt the hierarchical hypergraph structure to capture the semantic-level association of multiple regions from a global perspective. However, the construction of the hypergraph requires a more detailed description. Specifically, the definition of hyperedges and capsules in Section 4.3 needs to be clarified. The paper mentions that hyperedges are treated as high-level capsules, but what exactly does a capsule consist of? Furthermore, it would be helpful to explain the type of hypergraph connection that a hyperedge represents.

3. Some related work with similar ideas is not discussed. For example, the paper [1] also builds a pre-training model on urban data and builds a personalized model in a specific area. Though the idea is not the same, the author should discuss the difference in their related work at least.

4. It would be beneficial to include a brief discussion on future work in the paper.

[1] A Contextual Master-Slave Framework on Urban Region Graph for Urban Village Detection, ICDE 2023.


**Questions:**

1. What’s the “another initial” in Line 206 of Section 4.3.1 mean?

2. Is the aforementioned c in Section 4.4 from Section 4.3.1?


**Limitations:**

The authors have not adequately addressed the limitations.

---

> ### Author Rebuttal · Authors · 2023-08-10
>
> ***Response to Reviewer fFpq***
> **Comment 1:** Clarifications of notations and formulas.
> **Response:** We sincerely apologize for any confusion that may have arisen regarding the notations and equations, and greatly appreciate your feedback. In order to enhance the clarity of our work, we will provide detailed explanations for each of these issues and will supplement and update them in subsequent versions.
> (1) $\textbf{M}_{r,t}$ represents the mask operation for the $r$-th region in the $t$-th time slot. The first-level hyperedges denotes the final embeddings $\bar{\textbf{s}}\in\mathbb{R}^{H_S\times T\times d}$ obtained from Section 4.3.1. In detail, the designed hierarchical hypergraph neural architecture contains the hypergraph capsule clustering network (Section 4.3.1) and the classes aware hypergraph network (Section 4.3.2). We consider the former as the first-level and the latter as the second-level (also high-level).
>
> (2) For the time-dynamic encoding, GPST generates time-dynamic feature extraction parameters $\textbf{W}_t\in\mathbb{R}^{d\times d}, \textbf{b}_t\in\mathbb{R}^d$ using the temporal features $\textbf{D}_t\in\mathbb{R}^{d'}$. Formally, the customization is conducted by $\textbf{W}_t = \textbf{D}_t^\top \cdot \bar{\textbf{W}},~\textbf{b}_t = \textbf{D}_t^\top \cdot \bar{\textbf{b}}$, where $\bar{\textbf{W}}\in\mathbb{R}^{d'\times d\times d}, \bar{\textbf{b}}\in\mathbb{R}^{d'\times d}$ are learnable parameters for transformations and bias vectors.
>
> (3) $r$ and $t$ are the indexes of $R$ regions and $T$ time slots. For example, $\textbf{E}_{r,t}\in\mathbb{R}^d$ is the representation of the ST-data for the $r$-th region in the $t$-th time slot; $\textbf{E}_t\in\mathbb{R}^{R\times d}$ denotes the embedding matrix for all the $R$ regions in the $t$-th time slot; $\textbf{E}_r\in\mathbb{R}^{T\times d}$ denotes the embedding matrix for all the $T$ time slots in the $r$-th region.
>
> (4) The details of the loss function are presented in the supplementary material (Section A.1.3) for reference. Due to space constraints, we refrain from presenting it here again.
>
> (5) Another initial temporal embeddings $\textbf{D}_t^{'}$ is new temporal embeddings that  generated based on Equation (3). The aforementioned $\bar{c}$ in Section 4.4 is from Section 4.3.1.
>
> **Comment 2:** Further explanation of hypergraphs and capsule networks.
> **Response:** (1) We provide an explanation of the definition and construction of hypergraphs in Section 3 (Starting from line 114), as below:
>  A hypergraph $\mathcal{H} = \{\mathcal{V}, \mathcal{E}, \textbf{H}\}$ is composed of three parts: i) Vertices $\mathcal{V}=\{v_{r,t}: r\in R, t\in T\}$, each of which represents a region $r$ in a specific time slot $t$. ii) $H$ hyperedges $\mathcal{E}=\{e_1,...,e_H\}$, each of which connects multiple vertices to reflect the multipartite region-wise relations. iii)} The vertex-hyperedge connections $\textbf{H}\in\mathbb{R}^{N\times H}$, where $N$ denotes the number of vertices. To fully excavate the potential of hypergraphs in region-wise relation learning, we adopt a learnable hypergraph scheme where $\textbf{H}$ is derived from trainable parameters.
>
> (2) According to [1], a capsule is a group of neurons whose activity vector represents the instantiation parameters of a specific type of entity such as an object or an object part. We use the length of the activity vector to represent the probability that the entity exists and its orientation to represent the instantiation parameters. This characteristic empowers GPST with the ability to express region features, such as the functionality of a region. In contrast to low-level capsules, high-level capsules can represent the characteristics of more advanced entities, such as common features of a certain category of regions. By computing the agreement between low-level capsules and high-level capsules using a dynamic routing algorithm, the model can better capture the dynamic semantic information of regions.
>
> (3) According to the definition of hypergraphs, the hyperedges are not predefined but derived from trainable parameters. Therefore, each hyperedge in the hypergraph structure can represent a certain underlying semantic correlation. For example, in section 4.3.1, a hyperedge (also high-leval capsule) represents the probability of a region belonging to a certain category.
>
> [1] Dynamic Routing Between Capsule, NIPS 2017.
>
> **Comment 3:** The inclusion of the suggested work CMSF.
> **Response:** Thank you very much for your suggestion. We will incorporate it into the discussion of related work. Due to character limitations, we have included the discussion of all related works mentioned by the reviewers in the global response. Please refer to the author  rebuttal to access it.
>
> **Comment 4:** Further discussion of our future work for further investigation.
> **Response:** Thank you for your suggestions. We briefly mention future work in the supplementary material (Section A.4), and we will expand on it based on the latest developments.
> Future works：In future work, we will continue to explore more generalizable and versatile spatio-temporal pre-training frameworks. For example, one interesting research direction is addressing the issue of disparate distributions between training and testing data to enhance the practicality of pre-training models. Additionally, lightweight algorithms are also worth investigating, considering the high demand for computational efficiency in real-world urban scenarios.

---

> > ### Comment · Reviewer_fFpq · 2023-08-20
> > **Thanks for your response**
> >
> > Dear authors;
> >
> > Thanks for your effort to response my questions. I have no future points arising for the discussion.
> >
> > Regards

---

> > > ### Author Response · Authors · 2023-08-21
> > >
> > > We extend our sincere appreciation for your response.

---

### Official Review · Reviewer_H5wT · 2023-07-07

**Soundness:** 2 fair
**Presentation:** 2 fair
**Contribution:** 2 fair
**Rating:** 5
**Confidence:** 5

**Summary:**

This paper introduces a spatio-temporal pre-training framework called GPST (Generative Pre-training framework for Spatial Temporal prediction) to enhance the performance of existing models in traffic management and travel planning. The goal is to address the challenges of integrating and expanding refined models, while achieving better predictive performance.

The framework consists of two main components. (1) a spatio-temporal knowledge extractor that utilizes personalized parameter learners and hierarchical hypergraph networks. These modules are designed to model personalized representations and semantic relationships between regions, which have been overlooked in previous work. (2) an adaptive mask strategy is proposed to guide the knowledge extractor in learning robust spatio-temporal representations.

The paper conducts extensive experiments on representative benchmarks and demonstrates the effectiveness of the proposed GPST method.

**Strengths:**

1. The two research questions are interesting. Question (1) in personalized information extraction is well studied in STGNN. Question (2) is relatively novel since region functions could make a difference but not considered too much in the literature.
2. The experiments results look promising.

**Weaknesses:**

1. Related works should include SSL works from both Graph and Multivariate Time Series. I think current version has some missing references in esp. Graph SSL, which could be related to the strategy developed in this paper (needs further justification)
2. Figure 3 is really small to understand clearly. Visual illustration is better decoupled into text description.
3. Section 4 has too many details that could be moved to appendix since these detailed designs are not quite novel, though important. I'd suggest this section to be more compact and highlighted.
4. I have some concerns about the improvement by using GPST. It is clear in Table 1 that some methods benefit a lot and some not that much. I expect a more detailed analysis on this. Is some design of GPST redundant in GWN, MTGNN, etc? What is the core part of GPST that makes performance better? Or, is GPST dependent of certain architecture designs? How are these two factors (architecture and ssl strategy) decoupled?

**Questions:**

Thanks for clarifying the above mentioned weaknesses.

---

> ### Author Rebuttal · Authors · 2023-08-10
>
> ***Response to Reviewer H5wT***
> **Comment 1:** The inclusion of the suggested related work.
> **Response:** Thank you for your suggestion. The self-supervised learning methods for graph networks and multivariate time series forecasting will be incorporated into the related work in our revised version. Due to character limitations, we have included the discussion of all related works mentioned by the reviewers in the global response. Please refer to the author  rebuttal to access it.
>
> **Comment 2:** Explanation of figure size and corresponding text description.
> **Response:** We sincerely apologize for reducing the size of Figure 3 due to page limit. We will adjust it to its regular size in our revision. In addition, all textual descriptions for the figures will be added and updated. In instance, for Figure 1:  Upper left figure shows that the traffic patterns of same region in different time period behaves difference, while this situation also occurs in different regions during the same period (lower left); the middle figures show the relationship between regions changes dynamically over time (lower), while most of the existing works only construct static graphs (upper).The right figure shows the migration relationship between regions of different categories.
>
> **Comment 3:** Further discussion of the organization in Section 4.
> **Response:** Thank you for your valuable feedback. We appreciate your suggestions and will make the necessary adjustments to the structure and description of Section 4 to highlight the key points of our model. Specifically, to emphasize the key elements in each section, we will simplify the personalized parameter operations mentioned in Sections 4.2-4.4 by referencing their standard forms and moving their specific implementations to the supplementary material. Additionally, the specific details of the embedding layer mentioned in Section 4.2, which involve conventional operations, will also be included in the supplementary material. In Section 4.3, we will highlight the concept of Intra- & Inter-class Spatial Patterns Encoding and focus on describing the key module for modeling semantic correlations among multiple regions. The fundamental operations related to capsule networks will also be streamlined (e.g., Equation (6)).
>
> **Comment 4:** Further analysis of the performance improvement over the baseline GPST.
> **Response:** (1) Explanation of the performance improvement of GPST on different baselines. For the mentioned GWN and MTGNN, GPST shares some similar insights with these baselines, as all approaches adopt learnable modules to model spatial correlations. In GWN and MTGNN, learnable node embeddings are utilized to generate graph structures and learn node associations, while GPST employs a learnable hypergraph structure combined with capsule networks to capture region associations. Particularly, GPST considers the spatial-temporal semantic correlations in urban scenes, providing effective compensatory signals for GWN and MTGNN. Furthermore, experimental results demonstrate that with the inclusion of GPST, GWN and MTGNN exhibit faster loss reduction during training, as shown in the table below.
>
> | model/epoch | 5 |  10 |  20  |
> |:-----:|:-----:|:-----:|:-----:|
> | GWN | 19.79 | 18.67 | 17.14 |
> | w/ GPST |19.77 | 17.61 | 15.94 |
> | MTGNN | 17.84 | 16.75 | 16.21 |
> | w/ GPST |16.94 | 15.93 | 15.05 |
>
> (2) Does GPST rely solely on a specific module? From ablation study (Section 5.3), we observed that the personalized parameter learner, Intra- & Inter-class spatial patterns encoding and the adaptive mask strategy are all crucial modules in GPST, and they collectively contribute to its performance improvement. The injection of discriminative parameters can help the model to better capture the spatio-temporal relationship in different periods and different regions. Intra- & Inter-class spatial patterns encoding empowers the model with the ability to aggregate the information from regions with similar functionalities in the global perspective and discover inter-class correlation patterns. The adaptive mask further promotes Intra- & Inter-class association learning. These three components complement each other, contributing to the optimal performance achieved by GPST.
>
> (3) GPST exhibits decoupling and correlation in model architecture and training strategy. On one hand, model architecture and training strategy are distinct components with different responsibilities. The training strategy serves as a guide for the model architecture by employing masked signals to direct the learning process. The model architecture, acting as a feature generator, encodes complex spatio-temporal features driven by the training task. On the other hand, their common objective is to generate effective spatio-temporal representations. The proposed adaptive masking strategy is designed based on the model architecture, guiding the model to learn intra-class and inter-class associations in an easy-to-hard way, thereby generating higher-quality representations.

---

> > ### Comment · Reviewer_H5wT · 2023-08-21
> >
> > Thank the authors for reply. I don't have further questions.

---

> > > ### Author Response · Authors · 2023-08-21
> > >
> > > We sincerely appreciate your response.

---

### Author Rebuttal · Authors · 2023-08-10

***Responses Regarding Literature Review***
We extend our heartfelt gratitude to the reviewers for their valuable time and effort dedicated to evaluating our work. Due to constraints on character count, we will address the discussion of all relevant works mentioned by the reviewers in this section.

***Response to Reviewer H5wT***
**Comment 1:** The inclusion of the suggested related work.
**Response:** Thank you for your suggestion. The self-supervised learning methods for graph networks and multivariate time series forecasting will be incorporated into the related work in our revised version.
SSL methods for Graph and MTS: In recent years, self-supervised learning methods for graph have received significant attention in recent years. GNNs based on contrastive learning generate different views through data augmentation.  Following that, a loss function is employed to maximize the consistency of positive sample pairs while minimizing the consistency of negative sample pairs across views. For instance, GraphCL [1] generates two views of the original graph by applying node dropping and edge shuffling, and performs contrastive learning between them. Furthermore, JOAO [2] proposes an approach that automatically selects graph augmentations to facilitate contrastive learning. Another research direction involves generative graph neural networks, where these methods leverage the graph data itself as a natural supervisory signal and achieve representation learning through reconstruction. GPT-GNN [3] pretrains by reconstructing graph features and edges, while GraphMAE [4] employs node feature masking in both the graph encoder and decoder to reconstruct features and learn graph representations. In multivariate time series prediction, STEP [5] integrates long-term temporal features to generate temporal representations using a MAE-based pretraining method. COST [6] argues that unraveling temporal features is beneficial for time series prediction and proposes a contrastive learning approach applied to decouple trend and seasonal representations. However, spatio-temporal forecasting requires considering complex temporal evolution patterns and spatial correlation mechanisms simultaneously, and the pre-training paradigm for such tasks is still an area of exploration.

[1] Graph Contrastive Learning with Augmentations, NIPS 2020.
[2] JOAO: Graph Contrastive Learning Automated, ICML 2021.
[3] GPT-GNN: Generative Pre-Training of Graph Neural Networks, KDD 2020.
[4] GraphMAE: Self-Supervised Masked Graph Autoencoders, KDD 2022.
[5] Pre-training enhanced spatial-temporal graph neural network for multivariate time series forecasting, KDD 2022.
[6] CoST: Contrastive Learning of Disentangled Seasonal-Trend Representations for Time Series Forecasting, ICLR 2022.

***Response to Reviewer 8n4n***
**Comment 6:** Discussion of recent methods for temporal graph networks.
Thank you for your suggestion. In response to this valuable feedback, we will incorporate a dedicated paragraph in our revised version to provide a detailed analysis and discussion of the mentioned methods and their relevance to the field of spatio-temporal prediction scenarios.
Temporal graph networks serve as a similar research baseline aimed at reasoning about dynamic graph structures. CAWs [7] proposed a causal anonymous walks approach to inductively represent dynamic graph networks, specifically addressing interaction prediction for newly appearing nodes. PINT [8] introduced injective temporal message passing and relative positional features to achieve more expressive temporal graph networks. Additionally, GraphMixer [9] argued that complex model structures may not necessarily suitable for temporal networks and presented a conceptually and technically simple architecture based on MLP for link prediction.

[7] Inductive Representation Learning in Temporal Networks via Causal Anonymous Walks, ICLR 2021.
[8] Provably expressive temporal graph networks, NIPS 2022.
[9] Do We Really Need Complicated Model Architectures For Temporal Networks, ICLR 2023.

***Response to Reviewer fFpq***
**Comment 3:** The inclusion of the suggested work CMSF.
**Response:** Thank you very much for your suggestion. We will incorporate it into the discussion of related work.
CMSF [10] proposes a master-slave framework for identifying urban villages. The master model generates region representations in a pre-trained manner, while the slave model fine-tunes specific regions for accurate identification.
The proposed model can be distinguished from CMSF in the following three aspects.
(1) Task. CMSF focuses on identifying urban villages in the city, which falls under the classification task. In contrast, our model aim to predict the future spatio-temporal conditions given the historical records, thus belonging to a regression task.
(2) Technical approach. While CMSF employs techniques like GNNs to establish semantic correlations in static scenes, our task involves temporal dynamics. To address this, we have designed a hierarchical hypergraph framework with personalized parameter learner to model time-aware semantic correlations in regions. In addition, an adaptive masking strategy is proposed to further enhance the model's performance.
(3) Generalizability: After pre-training, CMSF requires specific configurations for the slave model to adapt to the master model's predictions, limiting its applicability to enhancing other similar models. Recognizing that different downstream baselines have distinct advantages and expressive capabilities when dealing with different data distributions, we have designed a universal pre-training framework that can enhance existing baselines to improve their performance.

[10] A Contextual Master-Slave Framework on Urban Region Graph for Urban Village Detection, ICDE 2023.

---

### Author Response · Authors · 2023-08-15
**General Response**

We extend our heartfelt appreciation to all the reviewers for their dedicated time and effort in reviewing this work. Their constructive feedback and valuable suggestions have greatly contributed to improving the paper.

We are delighted to hear that the reviewers find the problem addressed in our work to be interesting (Reviewer H5wT) and important (Reviewer 8n4n), and consider that the paper is well-presented and well-organized (Reviewer 83m7). We also appreciate the suggestions mentioned by the reviewers, such as enriching the related work (Reviewer H5wT, fFpq, 8n4n), highlighting key aspects of the methodology (Reviewer H5wT), including additional baselines and pre-training methods from similar works (Reviewer 8n4n), providing detailed descriptions of equations and notations (Reviewer fFpq), and improving the figure captions (Reviewer H5wT).

We have addressed each reviewer's comments with specific responses, and we will incorporate the necessary revisions in the paper to address their concerns. The key revisions are summarized below, and also mentioned in the corresponding responses to each reviewer.

(1) Related Work (Section 2): We have expanded this section based on the insightful suggestions provided by Reviewers H5wT, fFpq, and 8n4n.
(2) Methodology (Section 4): To enhance the conciseness and focus of the this section, we have condensed it and moved specific details of routine operations to the supplementary material, as recommended by Reviewer H5wT.
(3) Main Results (Section 5.2): Taking into account the valuable feedback from Reviewer 8n4n, we have included additional baseline results into the comparative experiments.
(4) Model Ablation Study (Section 5.3): In light of the insightful suggestion from Reviewer 8n4n, we have included some popular pre-training techniques of GNNs in the ablation experiments.
(5) Detailed Descriptions: Following the suggestions of Reviewer H5wT, we have supplemented the textual descriptions of Figures 1, 2, and 3. Additionally, in accordance with the suggestions from Reviewer fFpq, we have enriched some detailed descriptions of notations and relevant formulas.

We would like to express our gratitude once again to the reviewers for their valuable suggestions.

---

### Decision · Program_Chairs · 2023-09-21

**Decision:**

Accept (poster)

**Comment:**

The paper introduces GPST, a spatio-temporal pre-training framework designed to improve downstream prediction models. GPST incorporates personalization and region-wise semantic association in an unsupervised manner and utilizes a hierarchical hypergraph structure. Across four reviews, there is a consensus on the paper's innovative approach to spatio-temporal learning. The paper has been praised for its well-organized presentation, its ease of integration into existing models, and detailed experimental validation.

However, there are concerns across reviews about presentation clarity, scalability, and comparison with recent baselines. One reviewer questions the model's heavy reliance on the pre-training stage, suggesting that if the pre-training is not done adequately or the data used isn't representative, the model could underperform. Another reviewer suggests the inclusion of Points of Interest (POIs) for enhanced prediction performance.

In summary, Reviewers' ratings vary from borderline reject to accept, but leaning toward accept.